# Ancient DNA reveals admixture history and endogamy in the prehistoric Aegean

The Neolithic and Bronze Ages were highly transformative periods for the genetic history of Europe but for the Aegean—a region fundamental to Europe's prehistory—the biological dimensions of cultural transitions have been elucidated only to a limited extent so far. We have analysed newly generated genome-wide data from 102 ancient individuals from Crete, the Greek mainland and the Aegean Islands, spanning from the Neolithic to the Iron Age. We found that the early farmers from Crete shared the same ancestry as other contemporaneous Neolithic Aegeans. In contrast, the end of the Neolithic period and the following Early Bronze Age were marked by 'eastern' gene flow, which was predominantly of Anatolian origin in Crete. Confirming previous findings for additional Central/Eastern European ancestry in the Greek mainland by the Middle Bronze Age, we additionally show that such genetic signatures appeared in Crete gradually from the seventeenth to twelfth centuries BC, a period when the influence of the mainland over the island intensified. Biological and cultural connectedness within the Aegean is also supported by the finding of consanguineous endogamy practiced at high frequencies, unprecedented in the global ancient DNA record. Our results highlight the potential of archaeogenomic approaches in the Aegean for unravelling the interplay of genetic admixture, marital and other cultural practices.

The Aegean has long been recognized as a region of major importance for understanding transregional societal transformations between Europe and the Near East. Already during the seventh millennium BC, the first farming communities emerged in the Aegean, whereby the earliest evidence was unearthed on the island of Crete—that is, the oldest occupation level below the later palace of Knossos[1]—but the origins of these populations remain ambiguous. The next major transformation in Aegean prehistory took place during the Early Bronze Age (EBA; about 3100–2000 BC). Complex societies emerged, characterized by sophisticated architecture, metallurgy, sealing systems and the integration of the Aegean in the Bronze Age Eastern Mediterranean networks of exchange. During the late third millennium BC, the Greek mainland witnessed a severe societal breakdown (at the end of Early Helladic II) with lasting impact until the later Middle Helladic period of the early second millennium[2,3]. This disruption has been attributed to various factors, among them dramatic climatic changes[2,4,5] and the arrival of

new groups[6–8]. Crete does not seem to have suffered a comparable period of decline[9,10]. With the emergence of the first palaces during the nineteenth century BC in the Middle Minoan period, the island's societies transformed into a hitherto unknown sophistication in art, architecture and social practices.

Only a few centuries later, during the late Middle Bronze Age (MBA; Middle Helladic for the mainland), the first rich shaft graves of local elites appeared in southern mainland Greece, often displaying Minoan influences[11]. The competition between rising elites during the Shaft Grave period led to regional conflicts and culminated in the decline of many local dominions on the Greek mainland and possibly a first mainland military expedition to Crete during the fifteenth century[12]. At the end of this conflict, the palatial period (Late Helladic IIIA-B) started with a few eminent polities centred in Mycenae, Tiryns, Pylos, Athens, Hagios Vasileios in Laconia, Thebes, Orchomenos and Dimini—to name only the most prominent ones[13–15]. During this time,

✉e-mail: eirini_skourtanioti@eva.mpg.de; krause@eva.mpg.de; pajuccw@gmail.com; philipp.stockhammer@lmu.de

the influence on Crete by mainland centres intensified and Cretan resources were systematically exploited with the help of turning key palatial centres and cities like Knossos, Hagia Triada and Chania into outposts for the administration of large parts of the island[16]. So far, past human migrations in the Aegean were primarily reconstructed on the basis of archaeological and textual evidence but bioarchaeological studies have been adding new information during recent decades[17–22].

Biomolecular approaches based on ancient DNA (aDNA) have been introduced in prehistoric Aegean research during the last decade. The first aDNA study analysed mitochondrial genomes[23], emphasizing autochthonous developments rather than migration from outside Crete. Subsequent studies generated nuclear aDNA data and showed a common gene pool for the Aegean Neolithic populations, indicating that the southern Greek mainland differed from the northern in its higher genetic affinity to early Holocene populations from the Iran/Caucasus[24,25]. Others reported the presence of this 'eastern' (Iran/Caucasus-associated) genetic component in both Bronze Age (BA) Cretan (Minoan) and southern Greek mainland (Mycenaean) populations[26]. However, the last carried additional ancestry linked to the Western Eurasian Steppe herders (WES)[27,28] or Armenia. Recently, Clemente and colleagues expanded the sampling scope of the BA Aegean to the northern mainland and the Aegean Islands corroborating the previous findings but also reporting higher WES-related ancestry in MBA individuals from northern Greece[29].

Recent archaeogenetic studies outside the Aegean have engaged into integrating biological information as elements of the past social organization and structures[30–33], whereby it is necessary to acknowledge that relational identities are not determined only through biological kinship[34]. Most approaches to past kinship in the Aegean were based on morphometric and non-metric analyses[17,19,35] and first PCR-based studies were unsuccessful[36]. However, the potential of this line of evidence from the Aegean BA is outstanding due to the richness of collective burials as an expression and constitution of social belonging within local communities[37].

## Results

### The archaeogenetic dataset
Here, we generated new genome-wide data from 102 prehistoric individuals from Aegean Neolithic ($n = 6$), BA ($n = 95$), as well as Iron Age contexts (IA; $n = 1$) (Fig. 1 and Supplementary Note 1), thereby achieving a fourfold increase in sample size from previously published datasets. This sample, owing to the geographical and temporal distribution, enables us to address complex features of admixture history and other biological aspects interwoven into these prehistoric societies (for example, marital practices). Nea Styra on the island of Euboea and Lazarides on the island of Aegina add to the post-Neolithic sites included that date to the time before the debated disruption around 2200 BC (the end of Early Helladic II on the Greek mainland). The remaining individuals from the mainland and the islands are attributed to the Mycenaean culture of the Late Bronze Age (LBA) (Aidonia, Glyka Nera, Lazarides, Koukounaries, Mygdalia and Tiryns). Most of the data come from Crete (66 of 102 individuals), in a time transect that covers early phases of the Neolithic (Aposelemis; late seventh to early sixth millennia BC) and the BA (Hagios Charalambos–Early-Middle Minoan; Chania, Aposelemis and Krousonas–Late Minoan). With the exception of Aposelemis and XAN035 from Chania (about 1700–1450 BC), all other Late Minoan individuals date between about 1400 and 1100 BC (LMII–III). All the analysed skeletal remains from Nea Styra, Mygdalia, BA Aposelemis, Krousonas, Aidonia and Hagios Charalambos belonged to the same within-site collective burial context; for the latter, *Yersinia pestis* and *Salmonella enterica* were also recently detected[38]. Extracted aDNA was immortalized into genomic libraries, part of which were enriched for 1,233,013 ancestry-informative single nucleotide polymorphisms (SNPs) (1240K) (Methods) and sequencing data were evaluated for aDNA preservation and contamination (Supplementary Tables 1 and 2).

In our inferences for the Aegean individuals, we re-appraised all previously published contemporaneous individuals from this area[24–26,29] (Fig. 1). We also radiocarbon dated 43 of the skeletal remains that yielded genome-wide data (Supplementary Table 3; Methods).

### Transregional genetic entanglements of Aegean populations
To visualize genetic ancestry variation, we first performed a principal component analysis (PCA) on modern West Eurasian populations and projected onto the first two PCs the ancient individuals from the Aegean and nearby regions (Fig. 2). The six individuals from Neolithic Aposelemis cluster with other early European and Anatolian/Aegean farmers, suggesting that the gene pool of Neolithic Crete was linked to the broader Aegean during that period. After around two millennia, the EBA and MBA individuals show a substantial change in their PC coordinates, shifted along PC2 towards Early Holocene Iran/Caucasus and the descending Chalcolithic and BA Anatolians/ BA Caucasians. This shift does not seem uniform, as the five individuals from Nea Styra, who were buried together in the same shaft grave, show substantial genetic variation. Finally, the LBA individuals deviate from these earlier BA individuals towards BA Central and Eastern Europe, suggesting multiphased genetic shifts in the Aegean since the Neolithic.

To formally test whether the remarks from the PCA are consistent with diachronic gene-flow events, we used $f$-statistics of the form $f_4$ (Mbuti, Test; Anatolian farmers, Aegean) (Methods; Supplementary Note 2) that contrast the various Aegean groups with the Anatolian farmers east of the Aegean (Supplementary Table 4). Affinities with far-eastern groups like Neolithic Iran are traced for Neolithic Aposelemis (or APO004) but only reach significance levels ($\geq 3$ s.e. or $Z \geq 3$) on the EBA group from Nea Styra and then prevail for most of the later Aegean BA groups. However, the LBA ones additionally share alleles with contemporaneous or earlier (Mesolithic) populations from Central and Eastern Europe (for example, Eastern European hunter-gatherers: EEHG, Germany 'Corded_Ware', 'Russia_Samara_EBA_Yamnaya' and 'Russia_North_Caucasus'). In addition, evidence of admixture from these groups was confirmed with admixture $f_3$ test (Supplementary Table 5 and Supplementary Note 2).

### Neolithic to Early/Middle Bronze Age
Informed by the $f$-statistics, we explored formal admixture models using the software qpAdm (Methods; Supplementary Note 2). First, we tested a no-admixture model, which treated every individual as a sister group of Neolithic western Anatolia ('W. Anatolia_N') and then models by adding sequentially Neolithic Iran ('W. Iran N') and EEHG (Fig. 3). Substantial EEHG coefficients were fitted only on LBA and the two MBA individuals from the northern mainland ranging from around 5% to 25%, which explains why for some of them the simpler Anatolia + Iran Neolithic model was also adequate. Notably, Iran/Caucasus-related genetic influx was inferred in published individuals from the later Neolithic phases on the mainland (I2318, I709 and I3920; Peloponnese, around the fifth millenium BC)—but not earlier—as well as most of the EBA individuals from Euboea, Aegina and Koufonisia. Overall, the genetic heterogeneity among the Late Neolithic (LN) to EBA is not correlated with time alone, since within the Nea Styra grave male individuals carried substantially varying proportions of Iranian-related ancestry. By applying DATES on the LN and EBA individuals from the mainland and the islands (Methods), we obtained an average admixture date of around $4300 \pm 250$ BC (Supplementary Table 6), which is slightly younger when estimated from the Nea Styra individuals alone (about $3900 \pm 460$ BC). This variance in admixture dates also corroborates ongoing biological admixing with incoming individuals from the east of the Aegean following the establishment of the first Neolithic Aegean communities.

We further evaluated genetic heterogeneity with cladality tests using qpWave (Extended Data Fig. 1). Our results confirmed that various pairs within EBA Euboea, Aegina and Koufonisia were not cladal

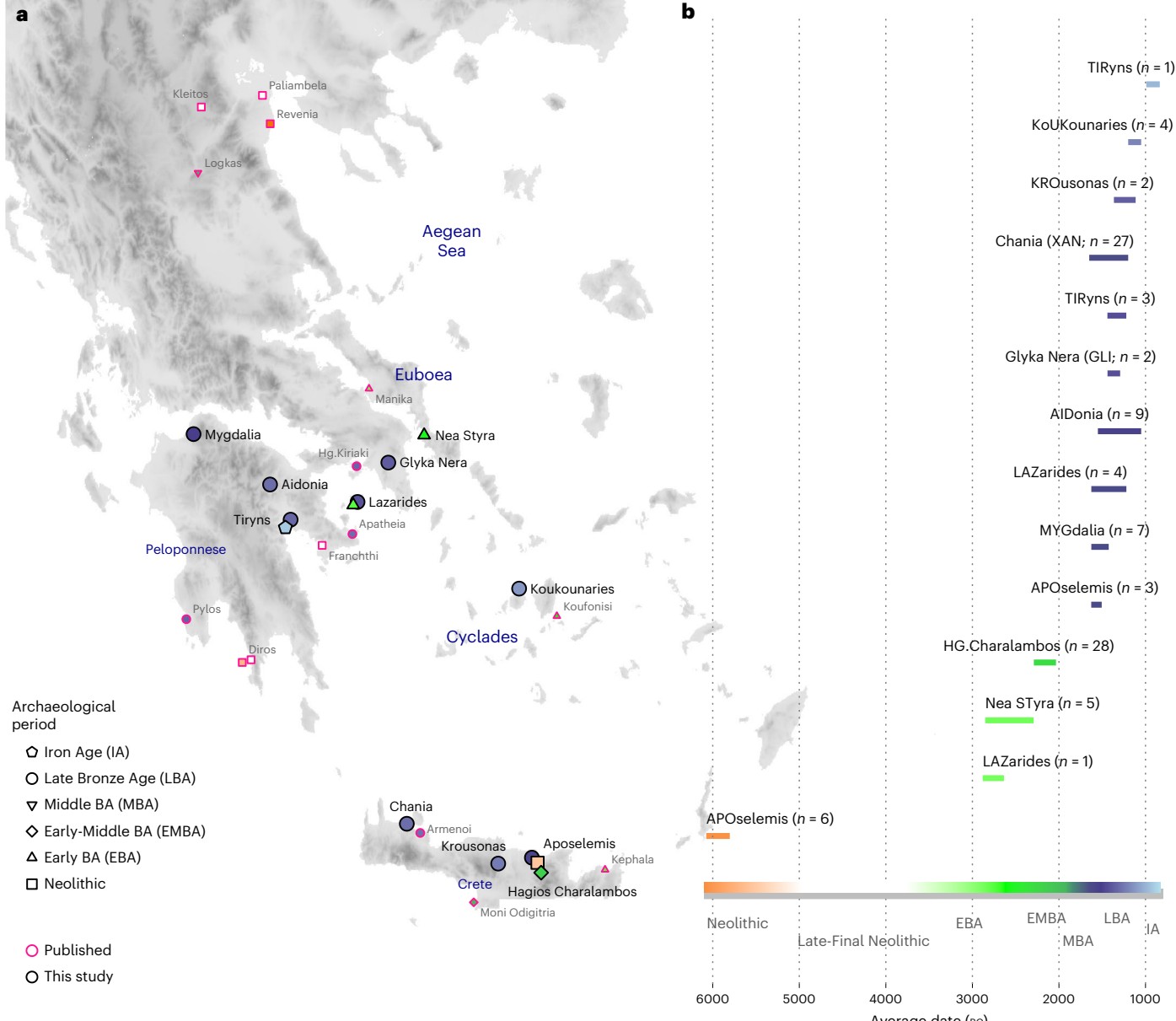

**Fig. 1 | Location and dates of individuals with newly generated aDNA data.**
**a**, Geographical distribution of archaeological sites mentioned in the study annotated by period. Sites with smaller symbols of light outline refer to the published datasets that are co-analysed and follow the same symbol/colour scheme. Data obtained from the same site but different periods, are annotated with jittering points. **b**, The number of individuals analysed and their date range based on archaeological chronology or radiocarbon dating. Site names are abbreviated in three-letter capitalized identifiers as indicated in the labels. E, Early; M, Middle; L, Late. See also Supplementary Tables 2 and 3.

to each other with respect to a set of reference populations (model $P < 0.05$), highlighting substantial genetic variation among coeval groups. In stark contrast, in Early Middle Bronze Age (EMBA) Crete the rate of non-cladal pairs (25 of 741) was the one expected for true models of cladal pairs to be rejected with a cutoff of 5% given a uniform distribution of the $P$ values.

To increase the resolution of admixture inferences, we repeated qpAdm in groups of individuals 'Crete Aposelemis N' ($n = 6$), 'S. Mainland-Islands LN-EBA' ($n = 13$) and 'Crete EMBA' ($n = 29$) following a 'competing' approach described in previous studies (Methods and Supplementary Information 2). For Aposelemis, the one-way model from Neolithic western Anatolia was adequate when aceramic farmers from central Anatolia (Boncuklu site) were included in the reference populations but the one-way model with the latter as a source failed

even without adding western Neolithic Anatolia to the references ($P = 9.32 \times 10^{-5}$) (Supplementary Note 2).

Subsequently, we modelled the differences of the two descending 'S. Mainland LN-EBA' and 'Crete EMBA' groups from the earlier Aegean farmers with two-way models from these local farmers and various southwest Asian populations (Supplementary Table 7). Most of the two-way models including Neolithic Aposelemis were not rejected, indicating a decreased resolution owing to the low SNP coverage and small group size of Aposelemis. On the contrary, when models included 'W. Anatolia N' as a local source instead, only the one with an additional 28% contribution from the Eneolithic/BA Southern Caucasus was feasible for 'S. Mainland-Islands LN-EBA' (Fig. 3b). Accordingly, for 'Crete EMBA', the additional ancestry was better modelled with Late Chalcolithic/ Early Bronze Age (LC-EBA) Anatolia (highest $P = 4.9 \times 10^{-3}$); however

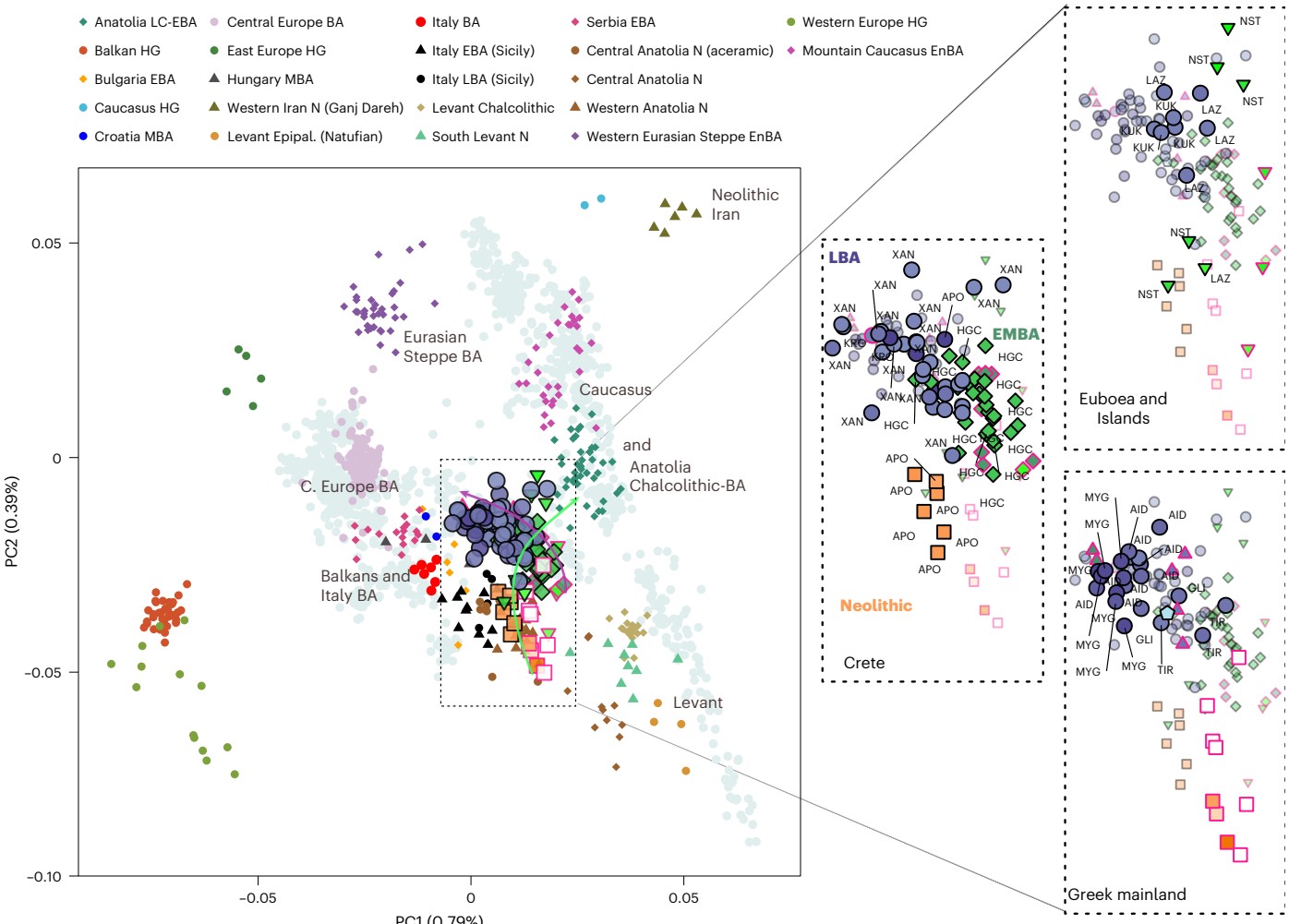

**Fig. 2 | West Eurasian PCA (grey background points) with projection of ancient Aegean and other ancient relevant populations (coloured points).** The arrows indicate the two major observed genetic shifts: from the Neolithic (N) to the EBA and from the MBA to the LBA. A zoom-in of coordinates for the Aegean samples is given and is subdivided by region (right). In every panel, the coordinates of the counterparts are plotted in the background in faded colours. The three-letter identifier of every individual is plotted as well. HG, hunter-gatherers; Epipal., Epipalaeolithic.

this model only became adequate as a three-way with an additional minute component (5%) from 'W. Iran N' (Fig. 3b).

### Mobility in the Middle/Late Bronze Age Aegean

For the LBA groups and the IA individual, we explored models of mixture from the corresponding ascending group ('S. Mainland-Islands LN-EBA' and 'Crete EMBA') and several European populations dated between around 3500 and 1000 BC (Supplementary Table 8). Informed by the previous analyses, we restricted the possible second sources to populations such as the EBA herders from the Pontic-Caspian Steppe (here grouped under 'W. Eurasian Steppe En-BA' and typically representing WES) and those shown to share a close genetic affinity with them. We first tested these models on 'Site_Period' groups, only if the cladality test (qpWave) agreed with grouping them as a homogeneous cluster (Supplementary Figs. 1 and 3a). Within the larger group from Chania, departures from cladality (*P* « 0.05) were more frequent (~10%) and were predominantly driven from specific individuals lying at the two ends of the EBA-LBA cline in the PCA (Extended Data Fig. 2b). To explore how these reflect significant differences in the admixture modelling, we analysed the group from Chania into the following three subgroups: 'Chania LBA (XAN030)', 'Chania LBA (a)' (XAN014, XAN028, XAN034) and 'Chania LBA (b)' (all the others) (Supplementary Table 8).

We found various sources ranging from East Europe, to Central and South Europe adequately fitting most models for the LBA Aegean groups. The smaller and heterogeneous sample of BA Bulgarian individuals or BA Sicily did not fit. Models with Serbia (EBA), Croatia (MBA) and Italy (EMBA) were adequate most of the time, while those with 'W. Eurasian Steppe En-BA' (En, Eneolithic) or some Central European source (for example, Germany LN-EBA 'Corded Ware') were adequate for all groups at the *P* ≥ 0.01 cutoff. Therefore, at the moment it is not possible to more closely identify the region(s) from where this genetic affinity was derived. Among the groups of the southern mainland, the estimated coefficients of the WES-related ancestry are overlapping (±1 s.e.) and average to 22.3% (Fig. 4a) but were substantially lower than for Logkas in the northern mainland (43–55% ± 4%). No significant differences were noted for IA Tiryns (±1 s.e.), indicating—albeit with limited evidence—genetic continuity after the end of the BA at least for this site. Similar coefficient ranges as in the southern mainland are observed for the nearby islands and the Cyclades, although the model for the one individual from Salamis shows no WES-related ancestry. In sharp contrast, in Crete, WES-related coefficients vary from 0% to about 40% clustering in three groups with significantly different coefficients. Among the individuals with minimal/no WES ancestry are the earliest, dating to the late seventeenth or sixteenth century BC Aposelemis, whereas the youngest (Krousonas, Armenoi; twelfth

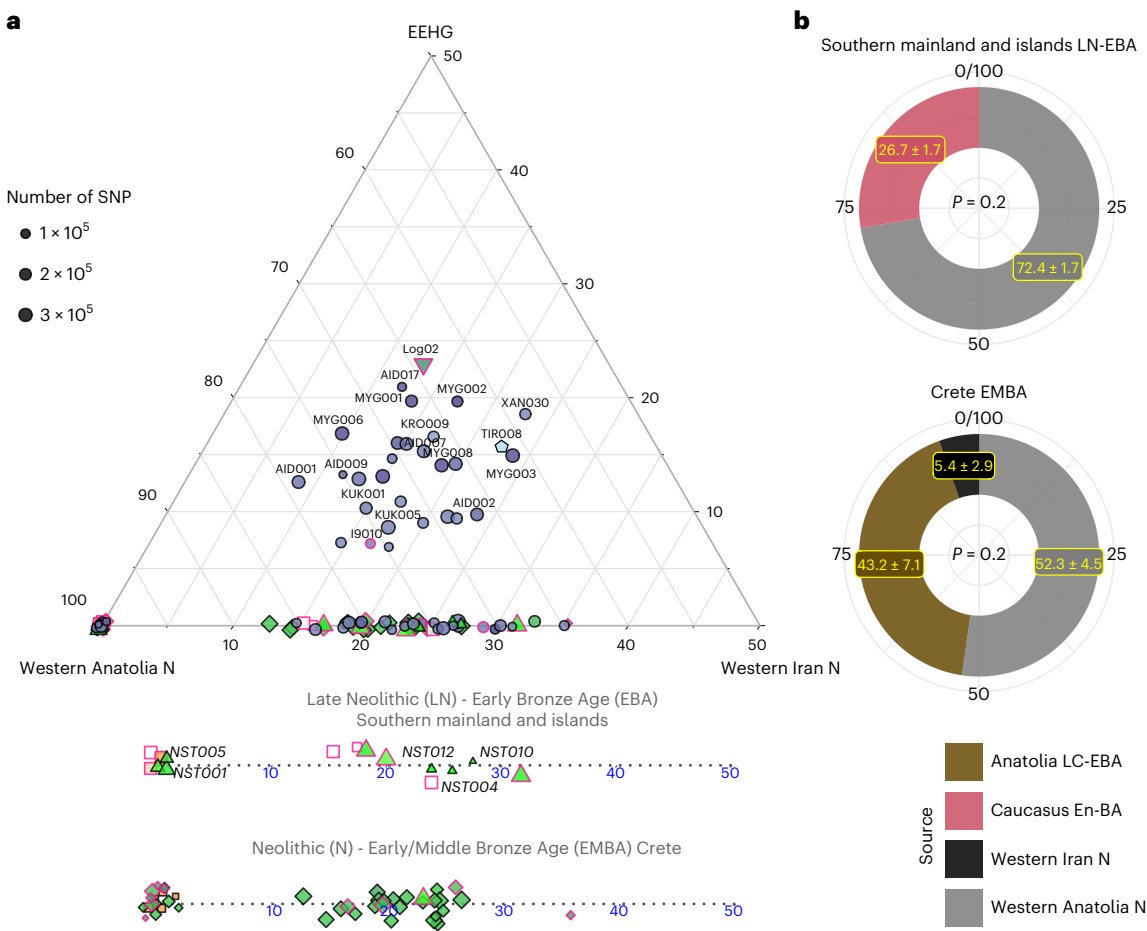

**Fig. 3 | Admixture modelling with qpAdm per individual and group. a**, Ternary plot for a three-way admixture model of Aegean individuals using the distal sources of ceramic farmers from Western Anatolia, Western Iranian farmers from Ganj Dareh and the EEHG, all dating to about 6000 BC. Because qpAdm is based on allele frequency differences, modelling of individual targets has a lower resolution especially when the SNP coverage is low. A few of the Late-Final Neolithic (LN) and EBA individuals show additional ancestry related to Neolithic Western Iran. To better visualize the fluctuation or Iranian-like coefficients among the LN-E/MBA individuals, the Anatolian–Iranian axis is also provided separately for Crete and the mainland islands. Fitting models were chosen with a cutoff of $P \geq 0.01$, with only four individuals falling in the lower range

($0.01 \leq P < 0.05$). **b**, Allele frequencies are averaged among all LN-EBA individuals from the southern mainland and all EMBA Cretan individuals and modelled using proximal in time and space source populations. For the former, a source proxy from the Eneolithic/BA Caucasus fits better than Anatolia, whereas the opposite holds for Crete. However, for the latter, the model becomes adequate with the inclusion of additional low contribution from Neolithic Iran. $P$ values and standard errors of mean were calculated by the qpAdm program applying a likelihood ratio test and the 5 cM block jackknifing method, respectively. No correction for multiple testing has been made. See also Extended Data Fig. 1 and Supplementary Tables 4–7.

century BC) harbour some of the highest amounts. However, within the ancient city of Chania, individuals spanning a short period of about three centuries display the entire range, a pattern consistent with an early phase of mixing between divergent populations.

To better understand these remarkable ancestry patterns in LBA Crete, we tested competing admixture models by interchanging the candidate second sources in which we now included 'Mainland MLBA' that consisted of all the individuals from the third panel of Fig. 4a (both southern and northern). For a comparison, we also tested the same models on the grouped targets 'Islands LBA' (Euboea, Aegina, Salamis and Cyclades), 'S. Mainland' and 'N. Mainland'—being aware that such artificial subdivisions of landscapes might not reflect past categorizations. The results are summarized in Fig. 4b. Interchanging the sources resulted in the rejection of some previously adequate sources (for example, 'Italy BA' for 'Islands LBA'), whereas models with Central or Eastern European sources remained adequate. However, two-way models with all of the above sources as well as 'Mainland MLBA'

fit the allele frequencies of all the LBA individuals from Crete ('Crete LBA'). This also applied when we modelled the two clusters from LBA Crete separately (Fig. 4a and Supplementary Table 9) but for the Crete LBA (group C) with high WES ancestry (individuals XAN030, KRO008, KRO009 and published Armenoi), just one source from 'Mainland MLBA' became adequate.

## Insights into sex bias, biological kinship and marital practices

Studies have shown that in some regions of Europe—like the Iberian Peninsula, Central Europe and Britain—the large-scale gene flow associated with the Eurasian Steppe during the BA resulted in the prevalence of the Y chromosome R1a and R1b haplogroups[28] or even involved male-biased admixture[33,39,40]. For the Aegean, we also estimated a significantly lower WES-ancestry proportion on the X chromosomes of the male individuals compared to most of the autosomes, which is consistent with male-biased admixture (Extended Data Fig. 3). However, only four out of the 30 male individuals dating post-sixteenth century BC (LBA and IA) carry the R1b1a1b Y haplogroup. The remaining—as well as the EBA/MBA ones—attest to the high prevalence of Y haplogroups J and G/G2

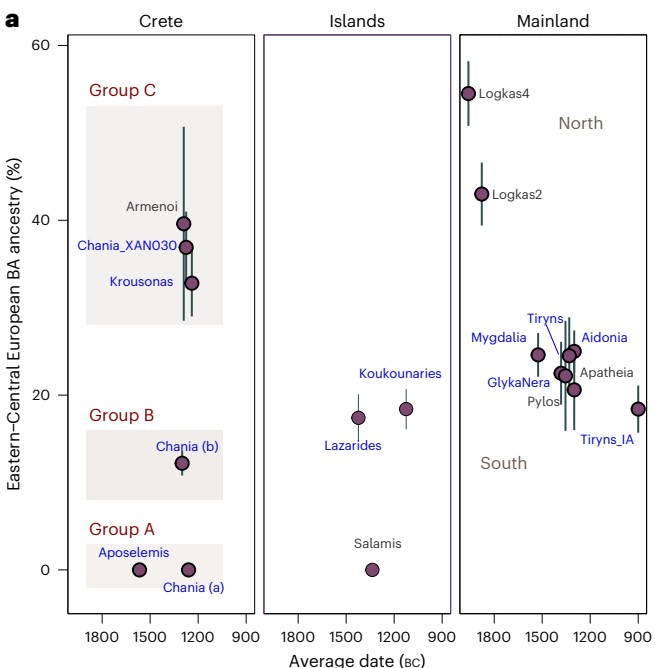

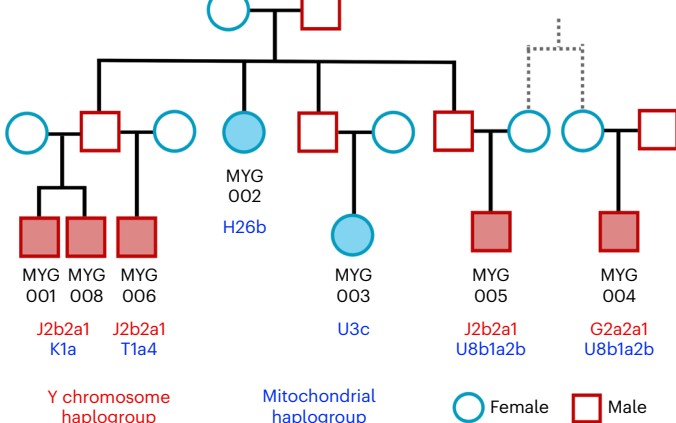

**Fig. 4 | Proximal two-way qpAdm models for the MLBA groups. a,** Estimated mean coefficient (coeff.) (±1 s.e.) of additional ancestry (WES-related) using as proxy a BA Central European population ('Germany LN–EBA Corded Ware'). For every group we assumed local ancestry in the models using the ascending population from the corresponding area (that is, EMBA Crete, LN-EBA southern Greek mainland and islands or LN northern mainland (for Logkas)). Newly reported LBA groups are annotated in blue letters. Before we applied the modelling on every 'Site_Period' group, we performed a test of cladality among all individuals which suggested substructure within the LBA site of Chania in Crete and resulted in three analysis groups. Overall, individuals from LBA Crete are distributed in three groups of non-overlapping WES-related ancestry estimations (A, B and C).

Models are supported with $P \geq 0.05$, with the exception of Tiryns_IA and Pylos with $P = 0.02$ and 0.04, respectively. **b,** Modelling results using the approach of rotating competing sources 2 in the right populations set (R11) (Supplementary Note 2) for Crete, the mainland and the islands. Low $P$ values (<0.01) indicate poor fit of the tested model and are annotated in red. For these models, the $P$ values are compared with the model fit without rotation of the sources. The gradual shift in Crete can be explained with admixture from the mainland but other proximal sources fit equally well. $P$ values and standard errors of mean were calculated by the qpAdm program applying a likelihood ratio test and the 5 cM block jackknifing method, respectively. No correction for multiple testing has been made. See also Extended Data Fig. 2 and Supplementary Tables 8 and 9.

(39 and 10 out of 59, respectively; Supplementary Table 2). These were already present in Early Holocene Iran/Caucasus and among Anatolian and European farmers[41–45] and very common in the Chalcolithic Anatolia and the Levant as well[42,46,47], further highlighting the importance of the contacts between the Aegean and southwest Asian populations since the Early Neolithic.

Biological relatedness and its representation in prehistoric collective burials has been poorly understood in the Aegean. Here, we present the first evidence for representation of biologically kin groups from a collective intramural infant grave dating to the LBA—a type of burial which existed since the Neolithic Aegean but became more common since the MBA[48,49]. Located within the Mycenaean (LBA) settlement in Mygdalia, a small cist grave was the primary inhumation of at least eight perinatal infants and one of the six child burials under the houses of the settlement (Supplementary Note 1). By estimating the degree of relatedness among seven of these infants (Methods; Extended Data Fig. 4 and Supplementary Note 3) and assigning the uniparental haplogroups (Supplementary Table 2), the relationship of the infants could be resolved in a single extended family tree whereby the six infants were the children and grandchildren of one couple (Fig. 5). The seventh individual (MYG004) was not a direct offspring of this family but related to MYG005 in the third degree through the maternal line, plausibly as first cousins.

Additional evidence of biological relatedness comes from Aidonia, where pairs of first- to third-degree relatives were determined among individuals buried within the three chamber tombs and the ossuary of Hagios Charalambos at the Lasithi plateau (Supplementary Note 1 and Extended Data Fig. 4). The individuals studied from

**Fig. 5 | Reconstruction of the family tree for the infants from the burial in Mygdalia (MYG; solid colour shapes).** The most parsimonious relationship between MYG004 and MYG005 is given. See also Extended Data Fig. 4.

Hagios Charalambos represent a secondary deposition of intermingled skeletons but were all unearthed from a particular section of the cave (Supplementary Note 1). Besides some pairs of close relatives (first to second degree), many pairs represent distant relatives. In addition to this high frequency of distant genetic relatedness, we also report extraordinarily high levels of consanguinity (~50% of the 27 individuals) estimated from the runs of homozygosity (ROH) by performing

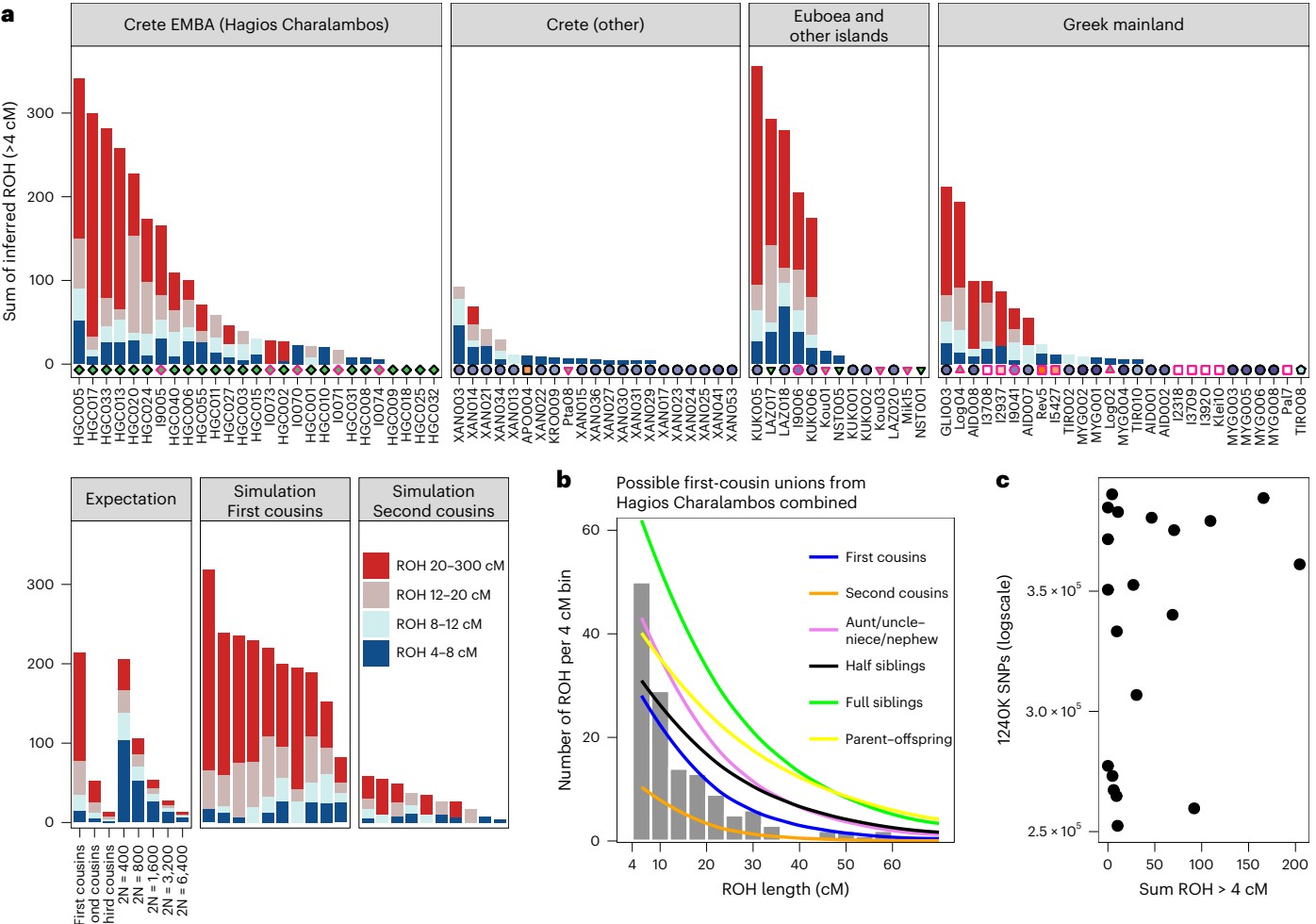

**Fig. 6 | Runs of homozygosity estimated with hapROH. a,** Inferred ROH per ancient Aegean individual. Results are plotted by area and the archaeological period/date of each individual is provided following the same symbol/colour scheme introduced in Fig. 1. Simulations and expectations for given parental relationships and demographic scenarios are given. For many individuals the ROH length distribution matches close-kin unions (first and second cousins).

**b,** Combined histogram of ROH length from all close-union offspring cases from the ossuary of Hagios Charalambos at the Lasithi plateau in Crete, compared to expected densities for certain parental relationships. See also Figs. 5 and 6. **c,** Scatterplot of lower coverage samples (250,000–400,000 SNPs) with total length of inferred ROH indicates that hapROH can reliably estimate long ROH at lower thresholds (Methods).

hapROH on the genotyping data[50] (Fig. 6a; Methods). The individual ROH histograms matched more with the expectations for parents being related to the degree of first cousins, half-siblings and aunt/uncle–nephew/niece (Extended Data Fig. 5). However, given the stochastic nature of genetic recombination and the often-compromised coverage of ancient samples, one individual's genome might only noisily match the expectations. Therefore, we combined the possible first-cousins unions cases and the cumulative histogram this produced favoured the parental relationship of first cousins against other scenarios (Fig. 6b and Extended Data Fig. 6). Coupling the evidence for frequent distant relatives and cousin–cousin unions suggests that those individuals formed a small endogamous community that regularly practiced first-cousin intermarriages.

Intriguingly, endogamy is not a unique feature of Hagios Charalambos. We applied the method on another 61 Aegean individuals from all the periods that met recommended SNP coverage thresholds. In total, we found that ~30% of the individuals have most of their ROH in the bin of the longest ROH blocks, consistent with being offspring of parents related to a degree equivalent to first and second cousins (Fig. 6a). Offspring of close-kin unions were identified from the Neolithic through the LBA but due to the uneven sampling no conclusions can be drawn

regarding temporal trends. Consanguinity was also present in higher frequency in the smaller islands of Salamis, Lazarides, Koukounaries and Koufonisia (50%) but overall it seemed common throughout the Aegean. The observed high frequency of endogamy diachronically points to a rather common social practice in the prehistoric Aegean that is so far unattested in the rest of the global aDNA record[50].

Finally, we observe a lowered genetic diversity among the Neolithic Aposelemis individuals, measured by a substantially reduced rate of mismatching alleles between pairs of samples (median $P_0 \approx 0.22$) (Extended Data Fig. 4; Methods). This signal can be due to several reasons. First, a lower $P_0$ would be consistent with Aposelemis being a small endogamous community; however the absence of any long ROH in APO004, the single individual with sufficient coverage to infer long ROH, does not support this hypothesis. Second, the lowered pairwise diversity could represent multiple pairs of second-degree relatives. However, to fit all pairs into a single consistent pedigree would require that all six individuals are half-siblings from either the maternal or the paternal side, with the exception of a single pair of full siblings (APO004–APO028). Due to the low SNP coverage in all the individuals, uniparental markers can neither rule out nor confirm such a pedigree but its high specificity places it as a less likely scenario.

Finally, long-term reduction of population size (bottlenecks) can cause lower population heterozygosity and such a signal has been previously reported for instance in hunter-gatherer groups and Cardial Neolithic Iberians[51,52]. Individuals from such drifted populations are expected to exhibit shorter ROH (4–8 cM), which are currently not detectable in low-coverage individuals such as APO004. Further supporting this scenario, the inferred heterozygosity (h) within the Aposelemis individuals was also reduced (mean h ≈ 0.1) and close to the expectation when assuming that the average pairwise diversity ($P_0$ ≈ 0.2) represents the diversity of the population and not pairs of close relatives. Summarizing, the current evidence is most consistent with the scenario of the Aposelemis early farmers descending from a small-sized population.

## Discussion

Our large-scale archaeogenomic approach provides new evidence regarding the role of human mobility in Aegean prehistory. The unprecedented finding of high frequency of consanguinity reveals a cultural practice otherwise unattested in the archaeological record.

First, our analyses on the Neolithic cemetery of Aposelemis, postdating the earliest levels at Knossos by about 1,000 years, suggest an Anatolian origin of the first Neolithic settlers, consistent with architectural, palaeobotanical and lithic evidence[53] and recent evaluation of wild and domestic fauna at those earliest levels that also suggest animal husbandry[54]. While a similar genetic connection was observed for coeval mainland populations[24,25], the genetic impact of Mesolithic and Neolithic populations from the other Aegean Islands, remain unknown but the evidence of a pre-Neolithic island horizon of a seafaring tradition[55] forces us to further elucidate the role of hunter-gatherers in the uptake of Neolithic subsistence practices in future studies. Thus, the reduced heterozygosity of the Aposelemis population might be interpreted as a coalescence of a small population of Anatolian farmers who settled the island in the early seventh millenium BC and remained biologically isolated for a period of time or as mobile small-sized populations arriving from nearby islands or a combination of both.

Subsequently, our findings indicate that the genetic landscape of Crete changed substantially since the early sixth millenium BC, marked by an influx of Anatolian populations inferred with our qpAdm modelling and admixture dating. Interestingly, eastern gene flow is also evident in other parts of Greece (Euboea, Aegina and Cyclades) since the LN but seems more episodic and oriented to populations from the Caucasus. In addition, although Y haplogroups are unresolved, male exogamy should be discussed as a plausible contributing factor to the heterogenous genetic profiles among the male individuals from Nea Styra, in line with evidence from biodistance on a neighbouring site[35]. Overall, while a more even sampling would be critical, current data seem to support that the eastern gene flow was not symmetric across the Aegean.

The disruption of life that is manifested in the Aegean and the Balkans via settlement dislocation during the late third millenium BC could be related to a breakdown of social structures and/or climatic challenges[56]. The finding of 'northern' ancestry in the MBA and LBA populations from the Greek mainland, does not support a large-scale population displacement but the north–south gradient indicates the directionality of this migration and population mingling. Some putatively proximal sources like 'Serbia EBA' or 'Bulgaria BA' failed to model this 'incoming' ancestry in many groups and R1b Y haplogroups were rather infrequent among LBA Aegean groups, all of which points to different migration dynamics in the BA Balkans and Greece, compared to other parts of Central and Western Europe.

A more direct demographic connection can be proposed regarding the LBA Cretan and Greek mainland populations. Following an horizon of destructions targeting palatial centres and elite symbols in Crete (Late Minoan IB)[57], material culture, funerary architecture and burial practices are believed to express innovations with features traditionally ascribed to the Mycenaean culture. On these grounds, an invasion of the island by people from the Greek mainland (around fifteenth century BC)

has been proposed but remains highly contested[12,58–60]. While unable to settle this debate decisively, the genetic analyses demonstrate that Cretan populations at larger port cities biologically mixed with populations coming to the island during the course of a few centuries. The presence of individuals with some of the highest WES-related ancestry proportions within LBA Aegean (Crete LBA group C), despite fitting with a scenario that the Greek mainland was the only source of incoming people, it could also suggest that populations from more distant areas (for example, Italy) contributed to the Crete LBA transition, a possibility that is supported in the material culture as well[61–63].

All different migrations proposed here (to Crete during the Neolithic and EBA, to the Greek mainland before the LBA and from the mainland to Crete during the LBA) differ in their bioarchaeological evidence, which, therefore, must not be seen as a simple proof of an archaeological hypothesis but as an additional perspective enabling us to unravel the complexity of past mobilities.

Finally, the evidence for consanguinity adds another layer regarding human mobility and social practices. Since the fundamental work by ref. 64, the phenomenon of cross-cousin unions has been much debated in anthropology, whereby in present-day societies, the evidence for cross-cousin unions is diverse, ranging from a common practice via toleration up to prohibition[65]. Different social, economic and ecological arguments have been brought forward as underlying reasons, for example, geographic isolation, endemic pathogen stress, integrity of inherited land and so on[66]. A combination of several factors combined with subsistence-specific needs (for example, olive cultivation forcing local constancy) might have shaped this practice in the Aegean. However, small population size was probably not a major reason in the Aegean as the reduced short-range ROH shown in our analyses is consistent with larger population sizes. Moreover, cross-cousin unions were practiced in different geographic contexts—on islands of different sizes as well as the Greek mainland and are not evident at some places during the second millennium (for example, Chania). Future studies need to further elucidate the factors that were responsible for the emergence, continuity and disappearance of marital practices. So far, the importance of cross-cousin unions in the prehistoric Aegean is unique among the currently available data for prehistoric endogamy, which is otherwise rarely evidenced[50,67–69]. This might indicate different standpoints with respect to marital practices of rural versus urban societies and/or that those were amenable to cultural influences and changed over time. Studying the interplay between past mortuary practices and social structure—including marital or residence rules—from an integrative bioarchaeological perspective has just become possible and future studies will help to refine our understanding of past social belonging.

## Methods

No statistical methods were applied for the determination of sample size and randomization.

The overall burial record from the Aegean Bronze Age is a corpus which underwent specific selection criteria in the past and has been subject to specific modes of preservation and excavation since then (for example, only individuals with a certain status and/or age were buried in a way that allows their study at present). The corpus of samples analysed in this study represents a broad variety of burial contexts (for example, shaft graves/collective graves, single graves, primary and secondary burials) through time and none of the burials would be termed 'elite' or 'outstanding' in its respective archaeological/historical context. There is also no sampling bias with respect to sex, age or locality of the burials and diverse cultural settings were included (for example, individuals from urban centres like Tiryns and Chania and remote hamlets like Mygdalia).

### Preparation of aDNA analysis

For the purpose of this study, we sampled 385 skeletal elements originally assigned to 357 ancient individuals. Teeth and petrous bones

made >95% of the sample corpus but when these elements were missing other parts such as tibia and femora were chosen. All sampling took place in a dedicated aDNA laboratory of MPI-SHH in Jena, following the laboratory's archived protocols https://doi.org/10.17504/protocols.io.bqebmtan and https://doi.org/10.17504/protocols.io.bdyvi7w6, the latter being an adaptation of a published protocol[70]. The aDNA extraction from most of the bone powder samples was performed with a modified silica-based protocol[71]. A detailed description of the steps is given in https://doi.org/10.17504/protocols.io.baksicwe. Genomic libraries were prepared from these extracts according to a double-stranded (ds) library protocol[72] with an initial step of partial UDG treatment[73] (https://doi.org/10.17504/protocols.io.bmh6k39e), followed by Illumina dual indexing (https://doi.org/10.17504/protocols.io.bakticwn). For a portion of the samples, we used an extraction-to-indexed library protocol supported by an automated liquid-handling system[74,75] which constructs libraries from single-stranded (ss) molecules. From every extract, at least one of the produced libraries was initially sequenced at a low depth (5–10 million reads) on an Illumina HiSeq400 platform with a setup of 50 cycles and paired-end or 75 cycles and single-read sequencing. Raw FastQC files were processed through EAGER pipeline[76] for removal of adaptors (AdapterRemoval v.2.2.0; ref. 77), mapping against the human reference hs37d5 with the Burrows–Wheeler aligner (BWA; v.0.7.12; ref. 78) with mapping quality and length filters of 30, and removal of PCR duplicates with dedup (v.0.12.2; ref. 76). Resulting information about library complexity and percentage of endogenous DNA was combined with mapDamage (v.2.0.6; ref. 79) estimates to evaluate the profile of endogenous aDNA preservation (Supplementary Table 1). Overall, our initial screening revealed that human aDNA preservation was very low to moderate (0.1–10% human endogenous DNA). Therefore, only aDNA enrichment methods are an economically viable strategy that allows one to generate data from a large number of individuals. Here, we chose to minimize batch effects and consistently generated in-solution hybridization enrichment data, consisting of ~1,2 million ancestry-informative positions (1240K capture)[28,43,80,81] from all samples with 0.1% human endogenous DNA or more. We note that a small proportion of the sampled libraries exhibited high DNA preservation (nine samples with >10% and up to ~40% endogenous content), which would make sequencing of the whole human genome cost-efficient and doing so could address additional research questions (for example, about rare variants). Only part of the immortalized libraries was used to produce enrichment data. The remaining libraries are permanently stored at the MPI-SHH/EVA laboratory facilities and future studies can use this resource to generate whole-genome data from these libraries.

Following the 1240K enrichment, the selected libraries were sequenced at standard ~20 million reads. For the evaluation of the post-1240K capture data, we rerun EAGER and mapDamage with the same settings. We also used the bed file of 1240K SNP positions to estimate on-target endogenous before-and-after 1240K capture and evaluate the performance of the protocol. We used Preseq (v.2.0; ref. 82) with the parameters <lc_extrap -s 1e5 -e 1e9> to predict the unique reads yielded in larger sequencing experiments. For libraries with low complexity, whenever that was possible, we opted for preparation of multiple libraries from the same extract. Additional sequencing data from the same library or multiple libraries from one DNA extract or same individual that were produced with the same protocols were processed equally and all data were merged at the level of bam files with Samtools (v.1.3) and dedup was run again. We authenticated aDNA using three different methods on the bam files that estimate modern DNA contamination on ancient samples. We analysed single-stranded, no-UDG-treated libraries with AuthentiCT (v.1.0.0; ref. 83) that relies on the distribution of damage-induced deamination across the length of the ancient molecules. We run the module for contamination estimate on males from ANGSD[84], which relies on heterozygosity on polymorphic SNPs on the X chromosome. We previously trimmed bams for terminal damage with trimBam (https://genome.sph.umich.edu/wiki/BamUtil:_trimBam) and reported the method 1 estimation. Finally, we analysed all libraries with schmutzi[85] after mapping mitochondrial reads with CircularMapper (v.1.93.5) and removing duplicates[76] and downsampling to 30,000 reads. Run modules contDeam and schmutzi estimated endogenous deamination, called an endogenous consensus and, based on this, computed the contamination rate. Ratios of mitochondrial/nuclear DNA that are very high (>200) can be unreliable for mitochondrial contamination estimates[86]. Therefore, when applicable, we relied on other methods and/or the behaviour of such samples in population genetic analyses.

The genetic sex was determined from a scatterplot of coverage on X and Y chromosomes normalized for autosomal coverage, which provided an unambiguous distinction between males and females and also matched the macroscopic estimations for adult individuals in all but a few exceptions (Supplementary Note 1).

We extracted genotypes from the pileups of original and trimmed bam files of ds libraries using the tool pileupCaller (https://github.com/stschiff/sequenceTools/tree/master/src/SequenceTools) and the option randomHaploid, which randomly chooses an allele to represent the genotype at a given SNP position. For the final genotype file, we kept transitions from the masked version and transversions from the original version. We genotyped the pileups from ss-library bams by activating the option singleStrandMode in pileupCaller which filters out forward-mapping reads with a C-T polymorphism and reverse-mapping reads with a G-A polymorphism, thereby effectively removing bias due to damage. Because of the differences in data production between ds and ss libraries, when applicable, we merged such libraries on the genotype level by randomly choosing a non-missing genotype at every position. Individuals with <20,000 SNPs, ≥10% contamination estimate or absence of such estimate were excluded from subsequent analyses. For selected individuals, we run pileupCaller with the option -randomDiploid and calculated within individual heterozygosity as number of ht sites/number of all sites.

We merged our final dataset with the release of publicly available genotype datasets of ancient and modern individuals (v.50.0) (https://reich.hms.harvard.edu/allen-ancient-dna-resource-aadr-downloadable-genotypes-present-day-and-ancient-dna-data), to which we added the recently published aDNA data from Italy[87] and based our inferences on a subset of the published data older than 2,000 years from across Eurasia. For the merging with the worldwide modern populations on the Human Origins array (~0.5 million SNPs) we kept the intersection of SNPs between the two panels. For downstream analyses we restricted all data to the 22 autosomes.

We assigned mitochondrial haplogroups and haplotypes from the consensus sequence (q30) generated by schmutzi and the software Haplogrep (v.2.1.25; ref. 88) applying a quality threshold of 0.65. To assign Y haplogroups, we filtered the pileup from the trimmed bams for ISOGG SNPs and for every such SNP we calculated its record of being either ancestral or derived. Then, via manual inspection we checked whether the presence of diagnostic SNPs for a given haplogroup followed a root-to-tip trajectory or whether there were spurious jumps in the phylogeny because of damage. For libraries with low coverage on mitochondrial and Y chromosome DNA, we additionally performed whole-genome and SNP enrichments, respectively, according to established protocols[81,89]. A summary of genetic sex, contamination estimates, SNP coverage and Y/mito-haplogroup assignments is given in Supplementary Table 2.

## Analysis of population structure

We performed PCA using the smartpca program from the EIGENSOFT (v.6.01) package[90]. To avoid bias in the calculation of PCs introduced by high rates of missingness on aDNA, we computed the PCA on 84 modern West Eurasian populations (1,264 individuals genotyped on the Illumina Affymetix Human Origins array) and projected ancient individuals with the option lsqproject.

## Admixture analysis with ADMIXTOOLS

We estimated $f$-statistics using the package ADMIXTOOLS (v.5.1; ref. 91). Depending on their formulation, $f$-statistics can provide a measure of genetic drift or test for hypotheses of admixture and allele sharing excess. While outgroup $f_3$-test of the form (Mbuti; X, Test)—for X and Test non-African populations—produces high values when X and Test share common drift, $f_4$(Mbuti, Y; X, Test) tests whether X and Y or Test and Y share more alleles than expected from the null hypothesis (X and Test cladal to Y). Therefore, $f_4$-statistics under given settings can provide useful hints for admixture and the possible sources. In addition, computation of $f_4$-statistics comes with a framework for block jackknife estimation of Z-scores, which we use for annotation of significant results ($|Z| \geq 3$). We also run admixture $f_3$(A; B, C) that tests whether the allele frequencies of population A are intermediate between B and C, with negative value indicating admixture. Using the information from the $f$-statistics results we built a framework for running tools qpWave and qpAdm from the same package. A detailed description of the machinery behind these tools is provided in ref. 28. In brief, the method harnesses information about allele frequency differences calculated by multiple $f_4$-statistics that relate a set of reference (right) populations with a set of targets (left) populations. Specifically, qpWave is used to estimate the minimum number of independent gene pools that explain a set of targets from the references. In practice, if two targets are related with the references as one gene pool, then they are cladal (undistinguishable) to the resolution of the references. In qpAdm, which is a derivative of qpWave, this principle is leveraged to model a target population as a mixture of contributions from $n$ source populations. The fit of the full model and the nested simpler models are evaluated and $P < 0.05$ or $0.01$ is generally interpreted as an inadequate explanation of the data. Admixture coefficients outside of the [0,1] range are also evidence of a poor fit of the full model. For the comparison of admixture coefficients from different chromosomes, we computed $Z = (\text{coefficient}_A - \text{coefficient}_X)/\sqrt{(\text{s.e.}_A^2 + \text{s.e.}_X^2)}$, where A was any of the 22 autosomes, X the sex chromosome X, s.e. the jackknife standard deviation from the qpAdm and applied a significance threshold of $Z \geq 3$.

To further discern differences in ancestries and their admixture coefficients by exploring source populations that potentially serve as proxies of the real sources in terms of time, space as well as the archaeological evidence, we applied a 'competing' approach described in previous studies[92,93]. In this approach, candidate source populations are interchanged between the reference (right) and source (left) populations in the qpAdm setting. If the one placed in the right population is a better proxy for the real source than the one tested in the left ones, the model is expected to fit poorly the data (low $P$ value).

## Admixture dating

We used the software DATES (v.753) (https://github.com/priyamoorjani/DATES) to test for exponential decay of local ancestry in a source population given two admixing sources. The decay rate is informative about the time since admixture; thus, the method can effectively date recent admixture events. A detailed explanation of the method is provided[47,94,95]. We run the method with standard parameters: in Morgan units binsize = 0.001 and fit of decay curve from 0.0045 (lovalfit) to 1 (maxdist) distance bins.

## Analysis of biological relatedness

For detection of closely related individuals, we applied the method READ[96]. In this approach, the coefficient of relatedness [0,1] between two individuals is estimated from their rate of mismatching allele ($P_0$) normalized with the pairwise allele differences among unrelated individuals within the population ($\alpha$), which is by default calculated as the median from the provided dataset. In this way, the method corrects for SNP ascertainment, marker density, genetic drift and inbreeding. An important implication from this formula is that for given $\alpha$, the $P_0$ for two identical individuals will be $\alpha/2$ and hence aDNA data from

samples belonging to the same individual can be easily detected. The method also calculates $P_0$ on non-overlapping windows of the genome and computes standard errors.

To detect relatives at a more distant degree, we run lcMLkin[97] on the masked versions of bam files with the options -l phred and -g best. This method uses a maximum likelihood framework to infer identical by descent (IBD) on low-coverage DNA sequencing data from genotype likelihoods computed with bcftools. The coefficient of relatedness $r$ is then calculated as $k_{1/2} + k_2$, with $k_1$ and $k_2$ the probabilities to share one or both alleles IBD, respectively. The method can also distinguish between parent–offspring ($k_0 = 0$) and siblings ($k_0 \geq 0$, depending on recombination rate) and in theory infer relatedness as distant as fifth degree. However, in low-quality data such as aDNA discrepancies from the expected $k_0, k_1, k_2$ values are common especially for comparisons relying on <10,000 SNPs[31].

To resolve pedigrees that differ in the IBD probabilities (for example, half-siblings or double first cousins), we performed gene imputations with GeneImp (v.1.3; ref. 98) and assessed matching and opposing homozygotes (Supplementary Note 3).

## Analysis of ROH

We inferred ROH using hapROH (v.1.0; ref. 50) (https://github.com/hringbauer/hapROH), a method designed to analyse low-coverage aDNA data by leveraging linkage disequilibrium from a panel of modern haplotype references. On 1240K data of at least 0.3× coverage, the method can successfully recover ROH longer than 4 cM. In cases of close parental relatedness, which produce long ROH in the offspring, the method can be efficient for detecting very long ROHs at an even lower coverage. Here, we called ROH in 65 of the Aegean samples (including previously published) with >250,000 SNPs. We simulated individual ROH for a given degree of parental relatedness using the software pedsim (https://github.com/williamslab/ped-sim) as described in Supplementary Section 4, hapROH. We used the embedded functions of the program for plotting the ROH as bars, individual or combined histograms and karyotypes.

## Direct AMS radiocarbon dating

Skeletal samples from 38 individuals were submitted to the radiocarbon dating facility of the Klaus-Tschira-Archäometrie-Zentrum at the CEZ Archaeometry gGmbH, Mannheim, Germany, which uses a MICADAS-AMS platform. The same sample from which DNA was extracted was preferred. Collagen was extracted from the bone samples, purified by ultrafiltration (fraction >30 kD) and freeze-dried. Collagen was combusted to $CO_2$ in an elemental analyser and $CO_2$ was converted catalytically to graphite. The $^{14}C$ ages were normalized to $\delta^{13}C = -25‰$ and were given in BP (before present, meaning years before 1950). The calibration was done using the datasets IntCal13 (ref. 99) and IntCal20 and the software SwissCal 1.0.

## Visualizations

We produced all graphs and maps with Rstudio (v.1.1.383), python (v.3.7) and Inkscape (v.0.92.4).

## Reporting summary

Further information on research design is available in the Nature Portfolio Reporting Summary linked to this article.

# Data availability

The raw (FASTQ) and aligned sequence data (BAM format; after MAPQ 30, length filter 30bp and removal of duplicates) reported in this paper can be accessed through the European Nucleotide Archive under the project name: PRJEB56216. Haploid genotype data for the 1240K panel are available in eigenstrat format (https://figshare.com/projects/Genotype_data_for_103_individuals_from_study_Ancient_DNA_reveals_admixture_history_and_endogamy_in_the_prehistoric_Aegean_/156152).

## Code availability

No new code and method were developed. Details on the settings for admixture modelling and dating are provided in Supplementary Note 2.

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

## Acknowledgements

We thank A. Mötsch for the support in the sample organization and management. We thank G. Brandt, A. Wissgott (MPI-Science of Human History; MPI-SHH) and the laboratory team from MPI-Evolutionary Anthropology for support in the laboratory work. We thank the members of the Max Planck-Harvard Research Center for the Archaeoscience of the ancient Mediterranean (MHAAM), the population genetics group from MPI-SHH/EVA, N. Patterson, M. Feldman, B. Rohrlach and K. Massy for their useful feedback. We warmly thank D. Michailidis and Z. Chalatsi from the Wiener Lab at the American School of Classical Studies at Athens for their support with sampling. We thank K. Prüfer for support in bioinformatics. We thank Arturo Cueva Temprana for support with the graphics. This study was funded by the Max Planck Society, the MHAAM, the National Research Foundation of Korea grant funded by the Korea government (2020R1C1C1003879) and the European Research Council (ERC) under the European Union's Horizon 2020 research innovation programme (ERC-2020-COG 101001951-MySocialBeIng) as part of P. W. Stockhammer's ERC Consolidator Grant project 'MySocialBeIng: Mycenaean Social Belonging from an Integrative Bioarchaeological Perspective'. We express our wholehearted gratitude to the late M. Kosma and the late D. Schilardi for their utmost contributions to Greek archaeology, as well as to years of research which contributed to making this study possible. They are both deeply missed.

## Author contributions

P.W.S., C.J., J.K. and E.S. conceived and supervised the study, and P.W.S., C.J. and J.K. acquired funding. P.W.S., A.A., M.A.-V., P.B., B.P.H., O.A.J., O.K., A.K., P.K., K.K., R.K., L.K., J.M., P.J.P.M., A. Papadimitriou, A. Papathanasiou, L.P.-M., K.P., N.P.-S., E.-A. Prevedorou, G.P., E. Protopapadaki, T.S.-S., M.S., K.S. and M.H.W. assembled, studied the archaeological material, supported sampling and/or advised on the archaeological background and interpretation of the results.

A.A., M.A.-V., P.B., B.P.H., O.A.J., A.K., E.K., R.K., L.K., J.M., P.J.P.M., A. Papathanasiou, L.P.-M., K.P., N.P.-S., S.P., E.-A. Prevedorou, G.P., E.P., M.S. and K.S. wrote the archaeological and sample background section. E.S., R.A.B., M.B., C.F., A.F., F.K., N.F.G.M., G.U.N. and A.T. performed laboratory work. E.S. and C.J. supervised the data analyses and E.S., H.R. and G.A.G.R. analysed the data. E.S., P.W.S., H.R. and C.J. wrote the manuscript with critical input from all co-authors.

## Funding

## Competing interests

The authors declare no competing interests.

## Additional information

**Extended data** is available for this paper at https://doi.org/10.1038/s41559-022-01952-3.

**Correspondence and requests for materials** should be addressed to Eirini Skourtanioti, Johannes Krause, Choongwon Jeong or Philipp W. Stockhammer.

Eirini Skourtanioti ®[1,2,3,30] ✉, Harald Ringbauer ®[1,2,4], Guido Alberto Gnecchi Ruscone ®[1,2,3], Raffaela Angelina Bianco[2,3], Marta Burri[2,3], Cäcilia Freund[2,3], Anja Furtwängler[1,2,3], Nuno Filipe Gomes Martins[2,3], Florian Knolle[2,3], Gunnar U. Neumann ®[1,2,3], Anthi Tiliakou[1,2,3], Anagnostis Agelarakis[5], Maria Andreadaki-Vlazaki[6], Philip Betancourt[7], Birgitta P. Hallager[8], Olivia A. Jones ®[9], Olga Kakavogianni[10], Athanasia Kanta[11], Panagiotis Karkanas[12], Efthymia Kataki[6], Konstantinos Kissas[13], Robert Koehl[14], Lynne Kvapil[15], Joseph Maran[16], Photini J. P. McGeorge[17], Alkestis Papadimitriou[18], Anastasia Papathanasiou[19], Lena Papazoglou-Manioudaki[20], Kostas Paschalidis[20], Naya Polychronakou-Sgouritsa[21], Sofia Preve[6], Eleni-Anna Prevedorou[12,22], Gypsy Price[23], Eftychia Protopapadaki[6], Tyede Schmidt-Schultz[24], Michael Schultz[24,25], Kim Shelton[26], Malcolm H. Wiener[27], Johannes Krause ®[1,2,3,30] ✉, Choongwon Jeong ®[28,30] ✉ & Philipp W. Stockhammer ®[1,2,3,29,30] ✉

[1]Department of Archaeogenetics, Max Planck Institute for Evolutionary Anthropology, Leipzig, Germany. [2]Max Planck Harvard Research Center for the Archaeoscience of the Ancient Mediterranean (MHAAM), Leipzig, Germany. [3]Department of Archaeogenetics, Max Planck Institute for the Science of Human History, Jena, Germany. [4]Department of Human Evolutionary Biology, Harvard University, Cambridge, MA, USA. [5]Department of History, Adelphi University, New York, NY, USA. [6]Ephorate of Antiquities of Chania, Hellenic Ministry of Culture and Sports, Chania, Greece. [7]Institute for Aegean Prehistory, Temple University, Philadelphia, PA, USA. [8]Danish Institute at Athens, Athens, Greece. [9]Department of Sociology and Anthropology, West Virginia University, Morgantown, WV, USA. [10]Ephorate of Antiquities of East Attica, Hellenic Ministry of Culture and Sports, Athens, Greece. [11]Antiquities for the Heraklion Prefecture (Director Emerita), Hellenic Ministry of Culture and Sports, Heraklion, Greece. [12]Malcolm H. Wiener Laboratory for Archaeological Science, American School of Classical Studies at Athens, Athens, Greece. [13]Ephorate of Antiquities of Arcadia, Hellenic Ministry of Culture and Sports, Tripoli, Greece. [14]Classical and Oriental Studies, Hunter College, New York, NY, USA. [15]Department of History, Anthropology, and Classics, Butler University, Indianapolis, IN, USA. [16]Institute for Prehistory, Protohistory and Near Eastern Archaeology, University of Heidelberg, Heidelberg, Germany. [17]British School at Athens, Athens, Greece. [18]Ephorate of Antiquities of Argolida, Hellenic Ministry of Culture and Sports, Nafplio, Greece. [19]Ephorate of Palaeoanthropology and Speleology, Hellenic Ministry of Culture and Sports, Athens, Greece. [20]National Archaeological Museum, Athens, Greece. [21]Department of Archaeology and History of Art, University of Athens, Athens, Greece. [22]School of Human Evolution and Social Change, Arizona State University, Tempe, AZ, USA. [23]SEARCH, Inc., Cornelius, NC, USA. [24]Center of Anatomy, University of Göttingen, Göttingen, Germany. [25]Department of Biology, University of Hildesheim, Hildesheim, Germany. [26]Department of Ancient Greek and Roman Studies, University of California, Berkeley, CA, USA. [27]Institute for Aegean Prehistory, Greenwich, CT, USA. [28]School of Biological Sciences, Seoul National University, Seoul, Republic of Korea. [29]Institute for Pre- and Protohistoric Archaeology and Archaeology of the Roman Provinces, Ludwig Maximilian University, Munich, Germany. [30]These authors jointly supervised this work: Philipp W. Stockhammer, Choongwon Jeong, Johannes Krause, Eirini Skourtanioti. ✉e-mail: eirini_skourtanioti@eva.mpg.de; krause@eva.mpg.de; pajuccw@gmail.com; philipp.stockhammer@lmu.de

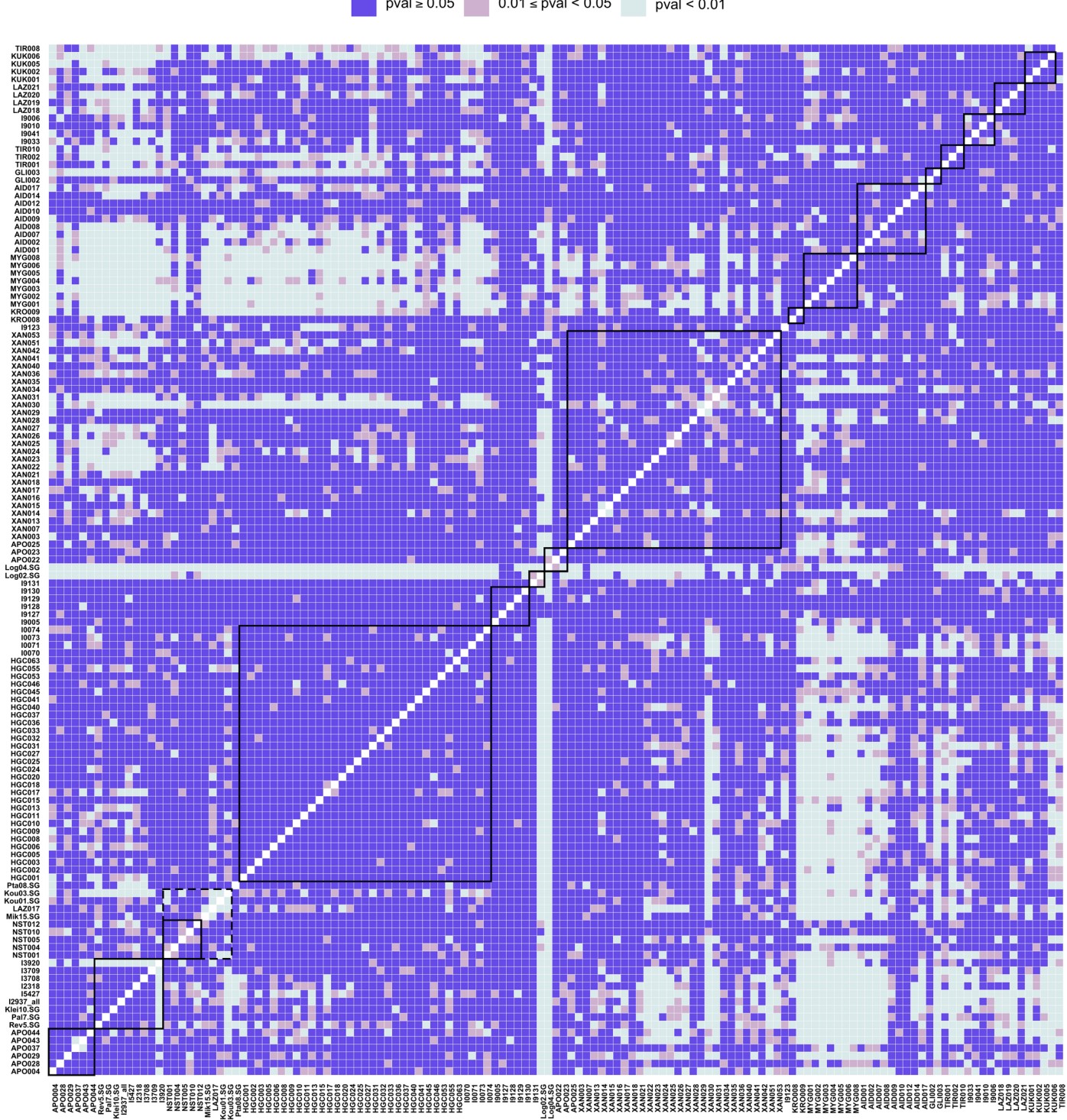

**Extended Data Fig. 1 | Heatmap of pairwise qpWave tests.** Low *P* values (conventionally < 0.05) are interpreted as a poor fit of the model and as more than one stream of ancestries being needed to explain the pair. Solid-line squares annotate clusters of individuals that date to the same period and come from the same archaeological site. Dashed-line square annotates Early Bronze Age (EBA) individuals from the islands of Euboea, Aegina and Koufonisia in Cyclades. Results are plotted in decreasing chronological order (Neolithic-Iron Age). We applied R11 (Supplementary Note 2) as a set of reference populations ('right pops'). *P* values were calculated by the qpWave program applying a likelihood ratio test. No correction for multiple testing was performed.

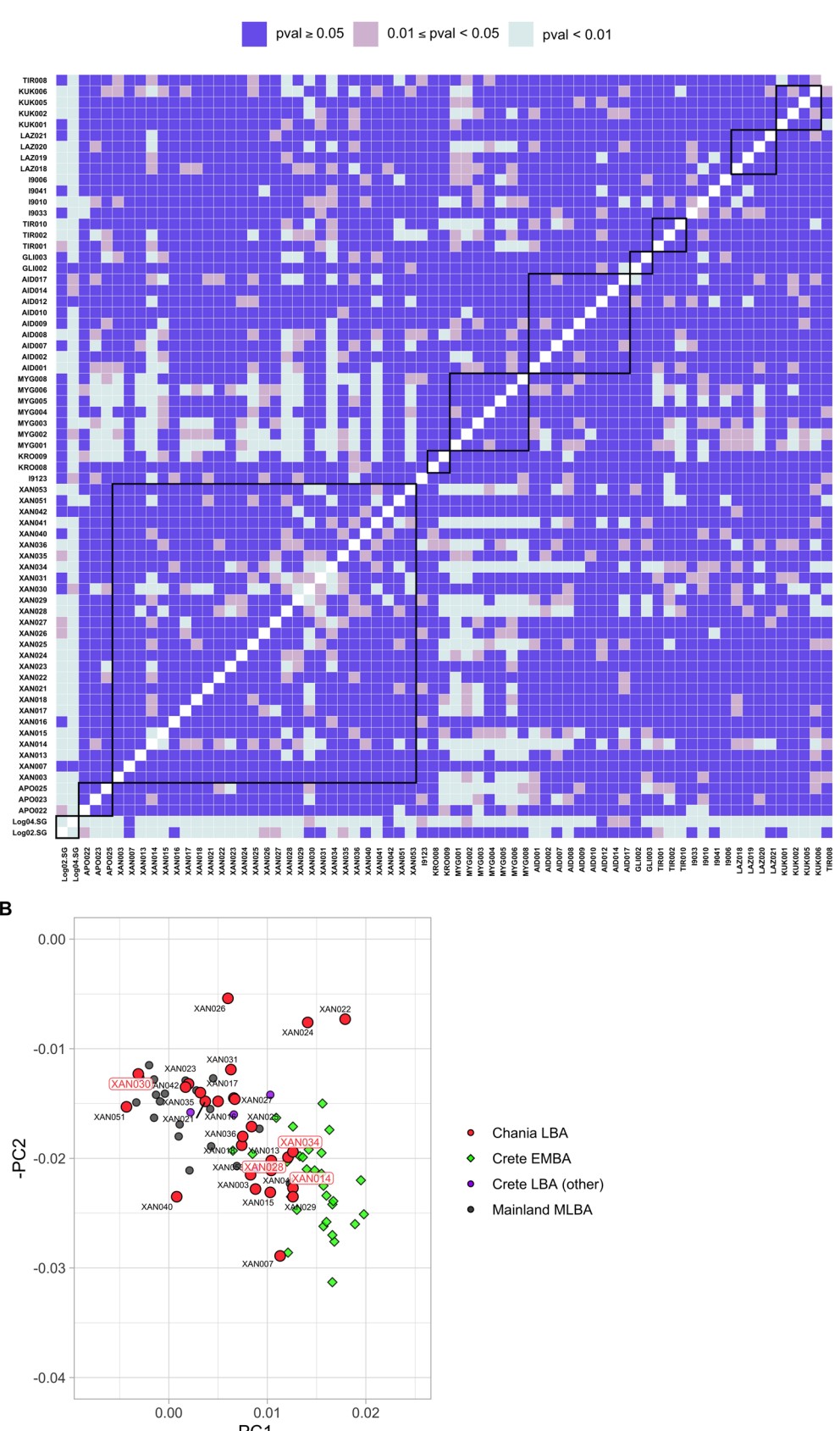

**Extended Data Fig. 2 | See next page for caption.**

**Extended Data Fig. 2 | Heatmap of pairwise qpWave tests and comparison with PCA coordinates. A**. Test of streams of ancestry necessary to explain a pair of individuals from a set of reference populations for the Middle/Late Bronze Age and one Iron Age individual from Tiryns. We repeated the analysis presented in Extended Data Fig. 1 by adding to the set of reference populations (R11) 'W. Eurasian Steppe En-BA'. This setting increased the rate of non-cladal pairs ($P <$ 0.01; at least two streams of ancestry) only among individuals from Chania (XAN) and led us to analyse Chania in three subgroups. $P$ values were calculated by the qpWave program applying a likelihood ratio test. No correction for multiple testing was performed. **B**. The PC1-PC2 coordinates from the Western Eurasian PCA displaying XAN individuals with their IDs. Those analysed separately are annotated in red letters (XAN014, XAN028 and XAN034 were grouped together and XAN030 apart).

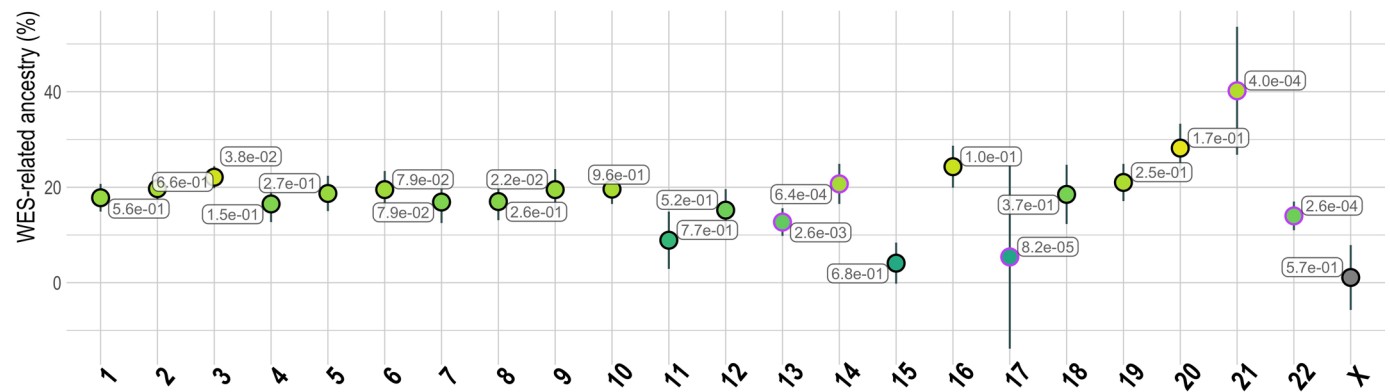

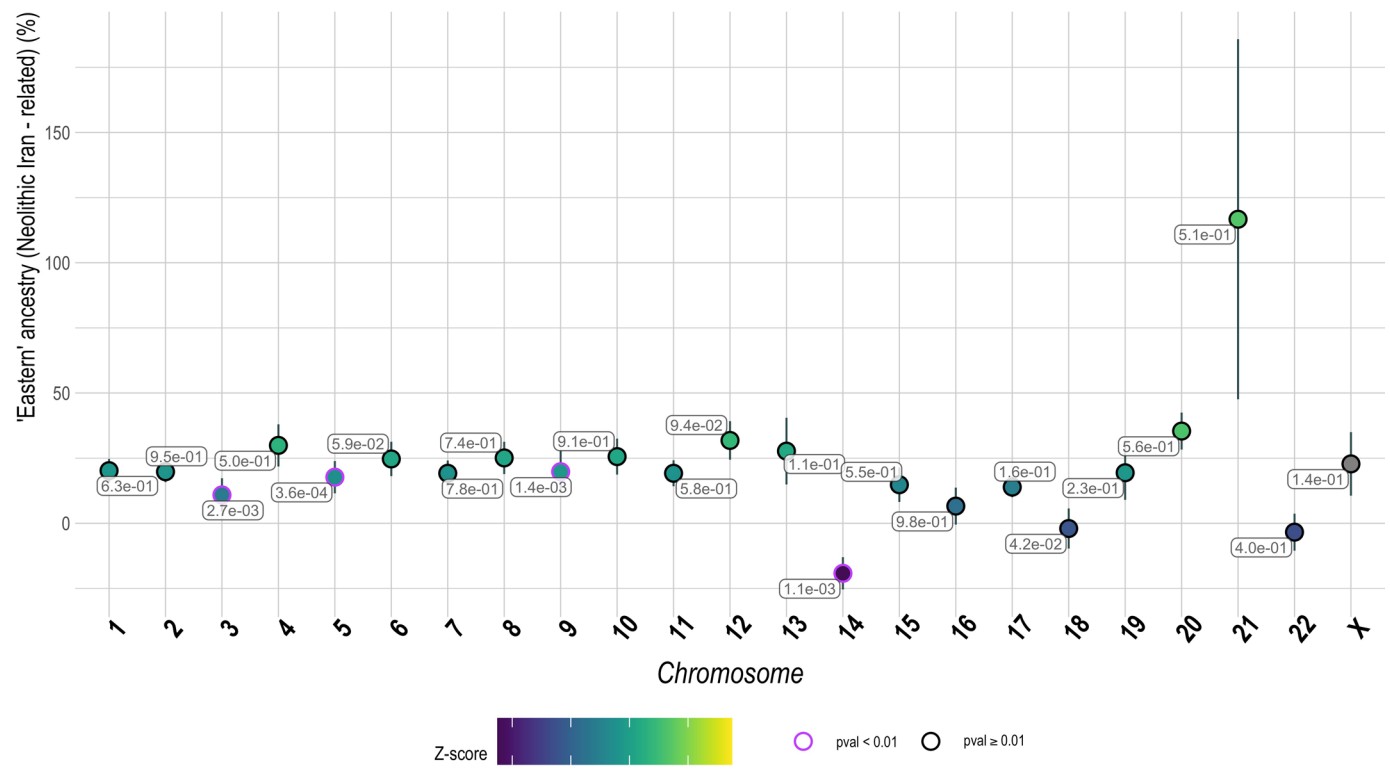

**Extended Data Fig. 3 | Estimated mean coefficients (±1SE) of additional post-Neolithic ancestries measured on all the autosomes separately, and the X chromosome of the Aegean male individuals grouped by period. A.** Positive coefficients from 'W. Eurasian Steppe En-BA' in LBA-IA males were fitted ($P \geq 0.01$) for most autosomes as well as chromosome X. WES-related ancestry estimated from the X chromosome was substantially lower compared to the autosomes, although only a few of these comparisons were significant (Z-score $\geq 3$). **B.** The same analysis for the 'eastern' ancestry indicates no sex bias in admixture between the Late Neolithic and the Middle Bronze Age. *P* values and standard errors of mean were calculated by the qpAdm program applying a likelihood ratio test and the 5 cM block jackknifing method, respectively. No correction for multiple testing was performed.

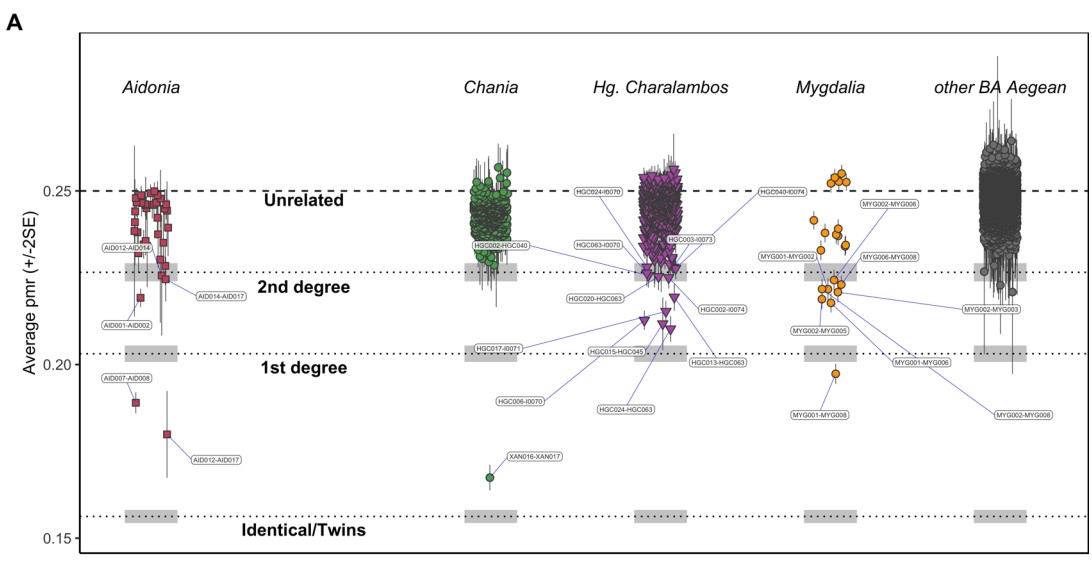

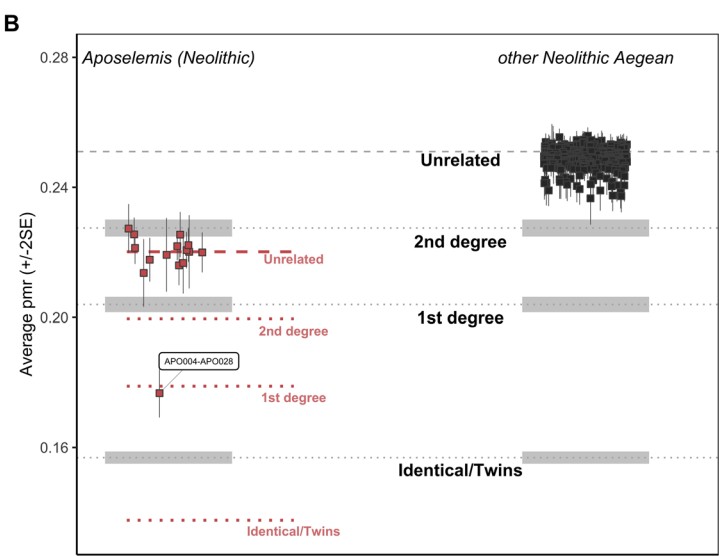

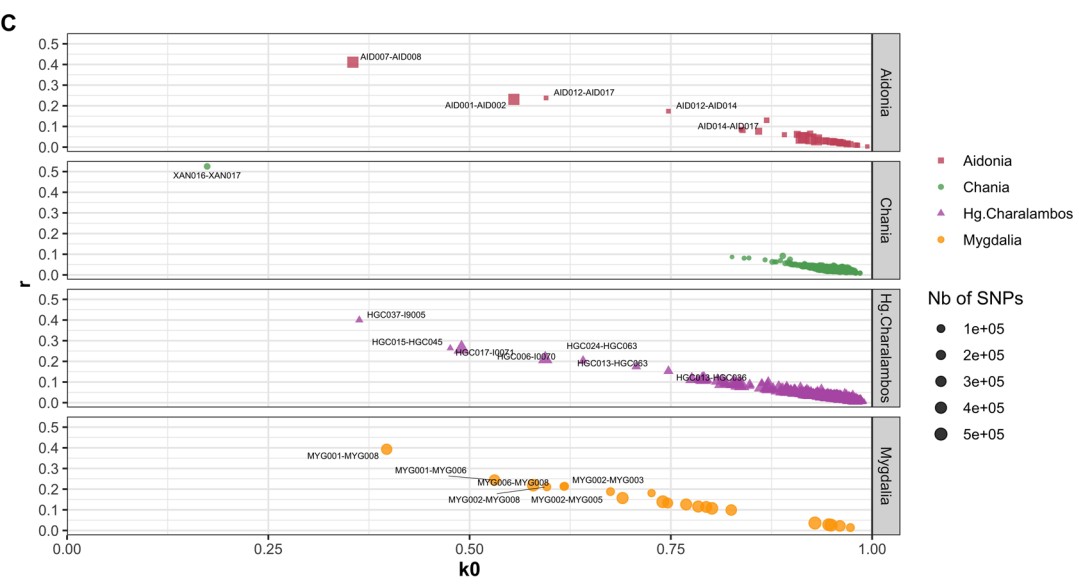

**Extended Data Fig. 4 | See next page for caption.**

**Extended Data Fig. 4 | Estimation of genetic relatedness with two different methods. A**. The pairwise differences (*PO*) were computed with READ and are plotted as ±2SE of the mean. The dashed line indicates the median value calculated from all pairwise comparisons used for normalization (baseline of unrelatedness). Dotted lines show the cutoffs for the classification to second and first degrees and identical/twins. Confidence intervals were calculated by the software and are indicated in gray shadows. Results are provided separately for sites with related individuals. **B**. READ results for Neolithic Aposelemis in comparison to other Aegean Neolithic sites from the Greek mainland and Western Anatolia (mean pairwise differences with ±2SE) suggest that the baseline of unrelatedness might be lower for the Aposelemis population, and normalization of *PO* produces lower cutoffs for close relatives (light-red lines). In this scenario, APO004 and APO028 are second-degree relatives. Because SNP ascertainment influences *PO* values, only individuals enriched for 1240K, or *in silico* genotyped on these SNPs were included. **C**. lcMLkin analysis. Scatterplot of *k0* against *r* for sites displaying pairs of relatives. First and up to third-degree relatives from Mygdalia are distinguished by both methods, as well as several pairs from Hagios Charalambos and Chania.

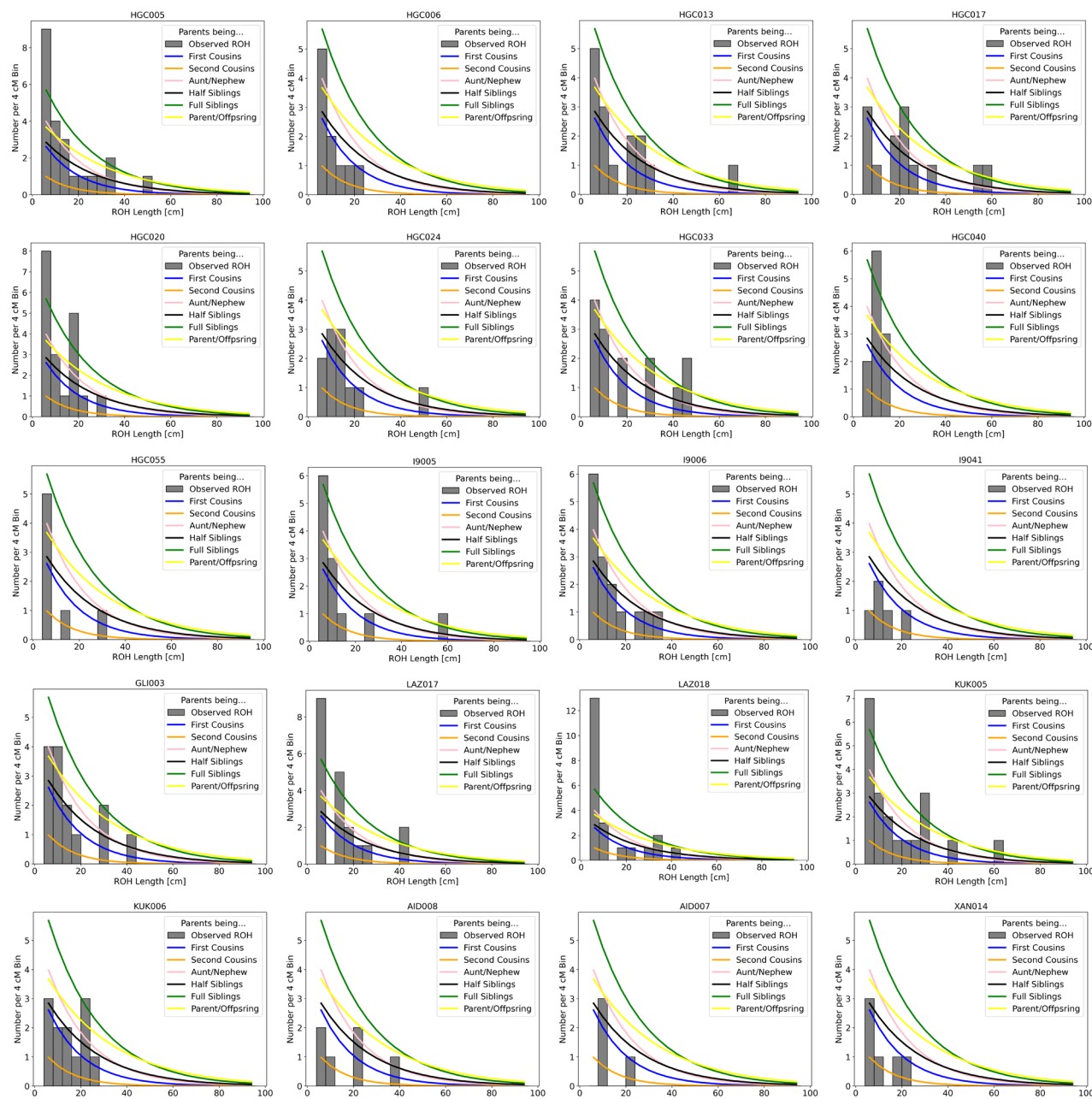

**Extended Data Fig. 5 | ROH length distribution for individuals with evidence of consanguinity (cross-cousin unions).** The ROH histograms are plotted for every case separately along with the expected densities for given parental relationships.

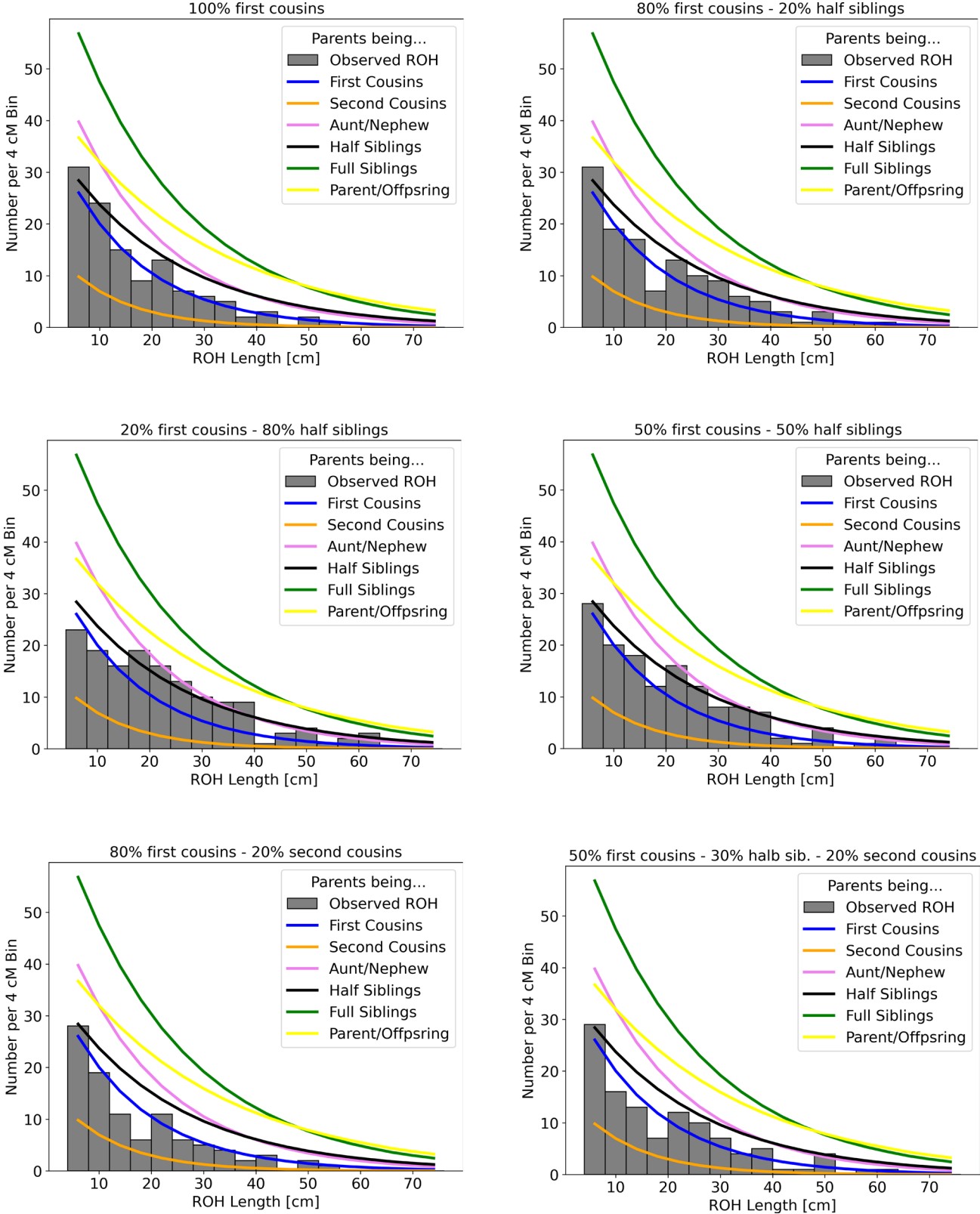

**Extended Data Fig. 6 | Histogram of ROH after combining simulated close-kin offspring, and expected densities for certain parental relationships.** For the three parental relatedness scenarios (half-siblings, first cousins and second cousins), 1000 offspring were simulated with the software pedsim (Methods). For comparison with Fig. 6b, the histogram of every panel was created after combining ten simulated individuals at different proportions. Histograms with all simulated first cousins, or 80% first cousins and 20% second cousins mostly closely match the histogram from the combined Hagios Charalambos individuals.

# Reporting Summary

## Statistics

For all statistical analyses, confirm that the following items are present in the figure legend, table legend, main text, or Methods section.

| n/a | Confirmed | |
|---|---|---|
| ☐ | ☒ | The exact sample size (*n*) for each experimental group/condition, given as a discrete number and unit of measurement |
| ☐ | ☒ | A statement on whether measurements were taken from distinct samples or whether the same sample was measured repeatedly |
| ☐ | ☒ | The statistical test(s) used AND whether they are one- or two-sided *Only common tests should be described solely by name; describe more complex techniques in the Methods section.* |
| ☐ | ☒ | A description of all covariates tested |
| ☐ | ☒ | A description of any assumptions or corrections, such as tests of normality and adjustment for multiple comparisons |
| ☐ | ☒ | A full description of the statistical parameters including central tendency (e.g. means) or other basic estimates (e.g. regression coefficient) AND variation (e.g. standard deviation) or associated estimates of uncertainty (e.g. confidence intervals) |
| ☐ | ☒ | For null hypothesis testing, the test statistic (e.g. *F*, *t*, *r*) with confidence intervals, effect sizes, degrees of freedom and *P* value noted *Give P values as exact values whenever suitable.* |
| ☒ | ☐ | For Bayesian analysis, information on the choice of priors and Markov chain Monte Carlo settings |
| ☒ | ☐ | For hierarchical and complex designs, identification of the appropriate level for tests and full reporting of outcomes |
| ☒ | ☐ | Estimates of effect sizes (e.g. Cohen's *d*, Pearson's *r*), indicating how they were calculated |

*Our web collection on statistics for biologists contains articles on many of the points above.*

## Software and code

Policy information about availability of computer code

| Data collection | No sofware was used for data collection (see section below). |
|---|---|
| Data analysis | EAGER (v1.92.59), AdapterRemoval (v2.2.0), BWA (v0.7.12), samtools (v1.3), dedup (v0.12.2), mapDamage (v2.0.6), Preseq (v2.0), CircularMapper (v1.93.5), Schmutzi, AuthentiCT(v1.0.0), ANGSD (v0.910), trimBam, pileupCaller, EIGENSOFT(7.2.1), ADMIXTOOLS (v5.1), DATES (v753), hapROH (v1.0), READ, lcMLkin, GATK (v3.5), GeneImp (v1.3), Haplogrep (v2.1.25) |

For manuscripts utilizing custom algorithms or software that are central to the research but not yet described in published literature, software must be made available to editors and reviewers. We strongly encourage code deposition in a community repository (e.g. GitHub). See the Nature Portfolio guidelines for submitting code & software for further information.

## Data

Policy information about availability of data

All manuscripts must include a data availability statement. This statement should provide the following information, where applicable:
- Accession codes, unique identifiers, or web links for publicly available datasets
- A description of any restrictions on data availability
- For clinical datasets or third party data, please ensure that the statement adheres to our policy

Sequencing data can be accessed through the European Nucleotide Archive (ENA) under project ID: PRJEB56216. Haploid genotype data for the 1240K panel in eigenstrat format will be also provided.

# Field-specific reporting

Please select the one below that is the best fit for your research. If you are not sure, read the appropriate sections before making your selection.

☐ Life sciences ☐ Behavioural & social sciences ☒ Ecological, evolutionary & environmental sciences

For a reference copy of the document with all sections, see nature.com/documents/nr-reporting-summary-flat.pdf

# Ecological, evolutionary & environmental sciences study design

All studies must disclose on these points even when the disclosure is negative.

| | |
|---|---|
| Study description | This study employs ancient DNA laboratory protocols to produce genome-wide data from human skeletal remains and applies statistical tools from the field of population genetics to address questions regarding the population history such as admixture, genetic relatedness, demography and consanguinity. |
| Research sample | Human skeletal remains from the Aegean (present-day Greece) that were recovered from archaeological excavations. |
| Sampling strategy | The overall burial record from the Aegean Neolithic and Bronze Age is a corpus which underwent specific selection criteria in the past and has been subject to specific modes of preservation and excavation since then (e.g., only individuals with a certain status and/or age were buried in a way that allows their study at present). The corpus of samples analyzed in this study represents a broad variety of burial contexts (e.g., shaft graves/collective graves, single graves, primary and secondary burials) through time, and comes from areas with distinct features with respect to their archaeological history. The majority of the samples dates to the Bronze Age, an archaeological period at the core of our questions regarding the contacts of the populations with neighbouring regions and the social organization. |
| Data collection | Bone powder was sampled following minimally invasive methods for sampling of archaeological material. DNA was extracted converted into a genomic library with adaptors for sequencing on Illumina platforms. |
| Timing and spatial scale | Timing scale: Neolithic (ca. 6000 BC; n=6), Bronze Age (ca. 2800-1050 BC; n=95), Iron Age (ca. 900 BC; n=1)<br>Spatial scale: Southern Greek mainland, Aegean islands and Crete. |
| Data exclusions | Processed samples for which a very low coverage of genetic markers was generated (e.g., ≤40,000 SNPs), or modern DNA contamination was estimated high. |
| Reproducibility | Sequence data will be uploaded to the European Nucleotide Archive, and all parameters (e.g., mapping quality filters, genotyping methods, admixtools) are provided in the Method's section and Supplementary Note 2. |
| Randomization | No statistical methods were applied for the determination of sample size and randomization. |
| Blinding | Blinding was not relevant/possible for our study, since all the data come from archaeological samples. |

Did the study involve field work? ☐ Yes ☒ No

# Reporting for specific materials, systems and methods

We require information from authors about some types of materials, experimental systems and methods used in many studies. Here, indicate whether each material, system or method listed is relevant to your study. If you are not sure if a list item applies to your research, read the appropriate section before selecting a response.

## Materials & experimental systems

| n/a | Involved in the study |
|---|---|
| ☒ | ☐ Antibodies |
| ☒ | ☐ Eukaryotic cell lines |
| ☐ | ☒ Palaeontology and archaeology |
| ☒ | ☐ Animals and other organisms |
| ☒ | ☐ Human research participants |
| ☒ | ☐ Clinical data |
| ☒ | ☐ Dual use research of concern |

## Methods

| n/a | Involved in the study |
|---|---|
| ☒ | ☐ ChIP-seq |
| ☒ | ☐ Flow cytometry |
| ☒ | ☐ MRI-based neuroimaging |

## Palaeontology and Archaeology

| | |
|---|---|
| Specimen provenance | Access to the material was granted through different applications: for every archaeological site a separate application was submitted to the Greek Ministry of Culture and Sports, and after its approval, skeletal samples and/or bone powder could be exported to Germany. All permits were issued in Greek and copies could be provided upon request. |

| Specimen deposition | Samples in the form of small fragments (e.g., petrous bones, teeth) or bone powder (max. 200 mg) were exported and sent to the Max Planck Institute for the Science of Human History (MPI-SHH) Lab facility, in Jena, Germany. |
|---|---|
| Dating methods | Radiocarbon dating with Accelerator Mass Spectrometry on bone/tooth samples weighing up to 1g. Samples were sent to the Klaus-Tschira-Archäometrie-Zentrum at the CEZ Archaeometry gGmbH, in Mannheim, Germany and were analyzed on a MICADAS-AMS platform. Measurements were calibrated using the datasets IntCal13 and IntCal20 and the software SwissCal 1.0. |

☒ Tick this box to confirm that the raw and calibrated dates are available in the paper or in Supplementary Information.

| Ethics oversight | No ethical approval/guidance was required. All the material was accessed after permission from the Greek Ministry of Culture and Sports and the agreement of the institutions/researchers who studied archaeologically the material and who also agreed to participate in this study. |
|---|---|

Note that full information on the approval of the study protocol must also be provided in the manuscript.

