## [Peer Review File · Nature Ecology & Evolution]

Peer Review Information

Journal: Nature Ecology & Evolution

Manuscript Title: Ancient DNA reveals admixture history and endogamy in the prehistoric Aegean

Corresponding author name(s): Eirini Skourtanioti

Editorial Notes:

Reviewer Comments & Decisions:

Decision Letter, initial version:

4th July 2022

Dear Ms Skourtanioti,

Your manuscript entitled "Ancient DNA reveals admixture history and endogamy in the prehistoric Aegean" has now been seen by the same three reviewers who were originally assigned to the manuscript at Nature, whose comments are attached. The reviewers have raised a number of concerns which will need to be addressed before we can offer publication in Nature Ecology & Evolution. We will therefore need to see your responses to the criticisms raised and to some editorial concerns, along with a revised manuscript, before we can reach a final decision regarding publication.

We therefore invite you to revise your manuscript taking into account all reviewer and editor comments. Please highlight all changes in the manuscript text file [OPTIONAL: in Microsoft Word format].

* If you have not done so already please begin to revise your manuscript so that it conforms to our Article format instructions at <http://www.nature.com/natecolevol/info/final-submission>. Refer also to any guidelines provided in this letter.

2[REDACTED]

Nature Ecology & Evolution is committed to improving transparency in authorship. As part of our efforts in this direction, we are now requesting that all authors identified as 'corresponding author' on published papers create and link their Open Researcher and Contributor Identifier (ORCID) with their account on the Manuscript Tracking System (MTS), prior to acceptance. ORCID helps the scientific community achieve unambiguous attribution of all scholarly contributions. You can create and link your ORCID from the home page of the MTS by clicking on 'Modify my Springer Nature account'. For more information please visit www.springernature.com/orcid.

[REDACTED]

Reviewer expertise:

as before

Reviewers' comments:

Reviewer #1 (Remarks to the Author):

A. Key results

The main conclusions reached in this article are the following:

(i) The paper reconstructs the genetic admixture history of NL to BA Aegean populations from the Greek mainland, the Aegean islands and Crete and postulates population movements to and from these regions.

(ii) The paper concludes that in the prehistoric Aegean practiced endogamy to an extended unprecedented in the ancient DNA record of Europe and Western Asia.

2B. Validity and significance

(i) The conclusions on the genetic history are valid and significant. The results (gene flow from the East in the EBA, connections to east Europeans in LBA mainland and adjacent islands, and a little bit later in Crete) confirm recent theories on mobility proposed by Aegean prehistorians. The conclusions are not original, but the paper does make a significant contribution in two ways: Genetic analyses provide us with a new set of independent data which can offer us a new perspective on past societies and increase spatiotemporal resolution if they are contextualized and interpreted correctly. The paper does this (at least in this part of the argument); the authors adopt an integrated approach which make important steps towards a better interpretation of genetic data. More on this below, under C. Validity of approach.

(ii) The conclusions on endogamy being practiced in NL to BA Aegean are novel and intriguing, but this part of the argument is less well developed. An interesting contrast is observed between the urban/palatial centre at Chania and the rural site at Hagios Charalambos, but no attempt is made to confirm this in other sites. Reference is made to Bintliff's (2017, 159) correlation between endogamy, social complexity and community size. However, Bintliff argues the opposite to what we observe at Chania – that endogamy is likely to arise when the size of a community increases beyond a set threshold. (It should perhaps be added that Bintliff's correlation is based on the kind of cross-cultural generalisations that characterized traditional evolutionary models now vehemently opposed by Graeber and Wengrow in *The Dawn of Everything*.) While the relation between demography and endogamy remains unclear, the discussion then attributes consanguinity to ecological factors which are not fully substantiated by the data, or cultural traditions which are not explored further. It is true that the relationship between mortuary practices and marital or residence rules in the past is new territory – but eventually geneticists (as well as prehistoric archaeologists) will need to delve more deeply into the theoretical aspects of kinship studies.

The results presented are of interest to scholars of (bio)archaeology, and the integrated approach adopted will increase its impact on the field.

C. Data & methodology

Validity of approach

Many recent genetic studies brought important refinements in analytical or computational techniques, but sometimes offered flawed interpretations because they failed to place aDNA results in their context. The integrated approach adopted in this study is clearly a step in the right direction: The authors do not over-privilege genetic data, avoid simplistic or sensational conclusions, and allow nuance and alternative interpretations. As an archaeologist, I would like even more attention to context. To give just one example: lumping together the mountain plateau of Lasithi in Crete, the island of Salamis, and a site in Euboea, which mediates between the mainland and island culture, under 'islands' is questionable. In general, the authors should be more aware that the analytical categories employed in this paper - 'the mainland', 'the islands' and 'Crete'- intersect with the terms 'Mycenaean' or 'Minoan' which collapse geographic, chronological, cultural, ethnic or political designations and represent fluctuating networks of connectivity.

See the remarks above (under B. Validity and significance) about part (ii) of the argument.

Quality of data

The authors stress that they have used the largest genetic sample from the Aegean, and indeed this is an important achievement.

The archaeological data, with all their contextual differences, have been taken more into consideration when interpreting the data. I maintain that more needs to be done in this direction: more attention needs to be paid into differences of context (site size and status in the settlement hierarchy, age/gender or status of burial population, social structure and cultural traditions). This is meant more as a recommendation for future research and eventual integration and synthesis than as an amendment of this paper.

Quality of presentation

The argument is built up in a clear and convincing way – although more so for part (i) than for part (ii) of the argument.

However, the end is abrupt and the two parts of the argument remain somewhat disjointed. They could be brought together more explicitly by stating that while the genetic history shows admixture and mobility from and to different directions, the postulated practice of consanguinity suggests, at least in some places and among some groups, a certain fixity in place and homogeneity in social practices across geographical, cultural, ethnic and political divides and through time. This could be done at the very end of the article.

D. Analytical approach

The statistical and computational methods employed fall outside the scope of my expertise.

E. Suggested improvements

The argument would improve

a. If the authors would be more aware that the spatiotemporal categories they use in part (i) of the argument are artificial, and that the networks of connectivity within which mobility should be understood fluctuate in size and scope.

b. If they would reflect more on part (ii) of the argument. it is advisable to avoid ecological explanations or vague references to cultural differences which have not been substantiated and to examine the theoretical premises of kinship structures more critically (the latter is meant more as recommendation for future research).

Both remarks can be addressed in the present paper by adjusting some formulations and without changing the scope of the argument.

F. Clarity and context: lucidity of abstract/summary, appropriateness of abstract, introduction and conclusions

- The abstract is clear and accessible. However, the phrase “these transitions have been elucidated only to a limited extent” disregards the huge archaeological bibliography on these transitional periods. It should be replaced by something like “the role of human mobility, and specifically of genetic admixture, during these transitions only now begins to be studied in earnest.”

- The introduction is clear.

- See above (C. Quality of presentation) for a suggestion to improve the conclusions.

G. References

Appropriate credit to previous work is given. It is good that references to studies which criticize biological determinism in genetic studies have been included (Brück 2021) although their message is still not fully internalized.

H. Minor remarks

- Why is Glyka Nera in the periphery of the Mycenaean world? (Supplementary Data, 1147-1148)?
- Clemente et al 2021 is not among the References

Sofia Voutsaki, Groningen

Reviewer #2 (Remarks to the Author):

Skourtanioti and colleagues have provided a thorough and thoughtful response to my comments. However, I have a few outstanding issues. These I believe are easily resolved and I'm happy to recommend for publication if addressed. I congratulate the authors on their excellent study, which I'm sure will draw much interest.

1. Methods and Data reporting

The authors briefly justify the lack of shotgun sequence to me in their response letter. However, it is important for the authors to provide these justifications to the reader in the main text, as well as the results from their initial shotgun screening in the supplementary tables. There are two reasons for this:

a) When destructive analysis is performed on archaeological material, there is an ethical imperative to get the most out of each individual sample. This is particularly true for time periods and geographical regions where DNA preservation is poor. As the authors note, data from only 30 ancient Aegean individuals was available prior to this study, and only a fraction of these are whole genome shotgun sequence. The authors had the opportunity to shotgun sequence at least seven of their samples (endogenous content >10%) to a coverage that would have more long-term value to the field but chose not to. I understand the logic underlying these decisions, but the authors should be up front with the reader about the trade-offs made and the limitations/advantages of capture data (e.g. the application of the same blanket approach to all samples, regardless of endogenous content, may help minimize batch effects).

b) The endogenous content results of the original shotgun screening may inform future genomic surveys of this region (e.g. which site types or geologies provide higher yields). Providing this data is also beneficial to the broader field as it provides important fodder for methodological improvement (which skeletal elements, extraction techniques etc. are providing best results).

Also, could the authors provide two more minor clarifications:

a) In their response letter they inform me that:

“For the vast majority of the libraries with 1240k SNP data, we initially performed shotgun metagenomic sequencing (ca. 5M reads)”

However, in the main text they state that:

“All libraries were initially sequenced at a low depth (5-10M reads) on an Illumina HiSeq400 platform”

Which statement is correct – the vast majority or all?

b) The use of the term “endogenous DNA” in the supplementary material is confusing, as it usually refers to the endogenous DNA content of an extract or library prior to any targeting techniques. The results of capture are typically reported in terms of capture efficiency and numbers of on-target and off-target reads. As I am requesting the authors to report the true endogenous content of each library from the screening data, could the authors update the terminology used for the post-capture data to avoid confusion.

2. Neolithic Affinities

I’m not quite satisfied with the authors response to my request for more dissection of Greek Neolithic affinities. Instead of carrying out further comparisons of Aposelemis_N and the Neolithic from the Peloponnese and Northern Mainland, they seem to have removed these two groups from most formal tests of admixture with the exception of Figure 3. For ease, I’ll refer to these populations as Mainland_N (excluding the youngest Peloponnese LN individuals with CHG-like ancestry likely derived from admixture in the 5th millennium BCE). This time around, I will be more direct with my requests for further analysis:

a) With regards to the PCA, perhaps my comments were not clear enough. Aposelemis_N does not plot with the Mainland_N samples. They differentiate on both PC1 and PC2 with W. Anatolians falling

6between the two groups. This may be driven by subtle differences in Levantine and Iranian ancestry. In Supp Table 4, Aposelemis_N shares a significant excess of alleles with Iran_C and near significant excesses with other populations high in Iranian_Neolithic ancestry. Why are the Mainland_N groups (S.Greece_EMNA and N.Greece_EN) not included in Supp Table 4? Do they show a different trend? Please add them and interpret the results.

b) Following from this, please perform direct comparisons between Aposelemis_N and Mainland_N using tests of the form $f_4(\text{Outgroup}, X_{\text{shotgun}}; \text{Aposelemis}_N, \text{Mainland}_N)$ or $f_4(\text{Outgroup}, X_{\text{capture}}; \text{Aposelemis}_N, \text{Mainland}_N_{\text{capture}})$ to avoid any confounding effects of capture bias.

c) Does a one-way model from Mainland_N also fit Aposelemis_N?

A minor comment: could the authors state within the main text that the genetic profile of the Mesolithic populations of the Aegean islands (Crete included) is unknown and it is possible that it was similar to that of Western Anatolian populations. There is no evidence of Villabruna-type ancestry any further southeast than the north Balkans (that I'm aware of?) and it possibly expanded from an Italian refugia. The general reader may not have this background context.

3. Minor corrections

Lines 205-207: This is hard to follow. Do you mean the earliest evidence of excess allele sharing with Neolithic Iran specifically is Nea Styra EBA? Or do you mean the earliest sample that gave a significant result for any test, which does not seem to be correct.

Figure 1: The color and shape key for archaeological sites sampled is a little confusing. Why do both IA and LBA have the same circle symbol? If the authors require more shapes try out the ggstar package (<https://cran.r-project.org/web/packages/ggstar/vignettes/ggstar.html>). Also, it is not immediately clear that multiple points have been plotted for the same site and perhaps jittered on the map (e.g. Tirnys, Lazarides, Aposelemis). Are multiple points also plotted for published sites? For example, Diros Cave has both Early and Late Neolithic samples but is colored as Early. Perhaps clarify this in the legend and explain the color key also. It would be useful if published sites were included in the time line.

Figure 3A: Please provide more of an explanation as to the meaning of the arrows and dotted axes placed below the main ternary plot. I think you are separating out Crete and Mainland samples along the Anatolia-Iran axis but it is not quite clear to reader.

Reviewer #3 (Remarks to the Author):

The authors have addressed in a satisfactory way virtually all the points I have raised in my previous evaluation of tehri manuscript, and I do not have any other major concern beside one.

Authors have mentioned in the text that HG.Charalambos have many related individuals. Maybe authors should explore how this increased relatedness is affecting the analyses based on this dataset as related individuals would be removed if the analysis would be done on modern day populations. Given the large set of individuals (n=28) I suspect that after pruning the sample size would be numerically not too different from the size of other groups they analysed (mostly in ranging between 1 and 9)

Minor concerns

In relation to the variation in Iran/Caucasus-related ancestry in the reply (and in teh ms) the authors mention that "male exogamy could also contribute to this pattern". It might be relevant here to verify if their Y chromosome/X chromosome data are compatible with this scenario. Whatever the situation (yes, or not/inconclusive), this should be mentioned.

In answering my question about relationship ebetween sample coverage and ROH authors refer to the methods and its development; my question was specific to this dataset: is there any relation between coverage and ROH in this dataset? Again, authors should include an appropriate comment in the ms.

When referring to the occurrence of cousin-cousin marriage authors refer to a wikipedia page to support their response. I would suggest to check the robustness of this by looking at more classical anthropological surveys and refer directly to them (as for example the Atlas of World Cultures by Murdock), also in the main text.

Faced with the options presented by authors, for clarity I would suggest to add a supplementary figure (in response to my comment on Figure 1 clarity).

I would suggest to split the Results section "Trans-regional genetic entanglements of Aegean populations" in subsections to make it more readable. AT the moment is almost four pages long; maybe a smaller section on Neolithic (which show little differences) and 2 on Bronze Age, maybe comparing continental/Crete results?

229-230: provide errors associated with these estimates

Occurrence of Y chromosome types (324-326): Occurrence of J and G chromosomes in Iran, Caucasus, Anatolia and Levant compatible with their other data linking these samples with these areas. Were these haplogroups present in other regions too and therefore similalry compatible?

*****END*****

Author Rebuttal to Initial comments

Reviewer expertise:

as before

Reviewers' comments:

Reviewer #1 (Remarks to the Author):

A. Key results

The main conclusions reached in this article are the following:

- (i) The paper reconstructs the genetic admixture history of NL to BA Aegean populations from the Greek mainland, the Aegean islands and Crete and postulates population movements to and from these regions.
- (ii) The paper concludes that in the prehistoric Aegean practiced endogamy to an extended unprecedented in the ancient DNA record of Europe and Western Asia.

B. Validity and significance

(i) The conclusions on the genetic history are valid and significant. The results (gene flow from the East in the EBA, connections to east Europeans in LBA mainland and adjacent islands, and a little bit later in Crete) confirm recent theories on mobility proposed by Aegean prehistorians. The conclusions are not original, but the paper does make a significant contribution in two ways: Genetic analyses provide us with a new set of independent data which can offer us a new perspective on past societies and increase spatiotemporal resolution if they are contextualized and interpreted correctly. The paper does this (at least in this part of the argument); the authors adopt an integrated approach which make important steps towards a better interpretation of genetic data. More on this below, under C. Validity of approach.

(ii) The conclusions on endogamy being practiced in NL to BA Aegean are novel and intriguing, but this part of the argument is less well developed. An interesting contrast is observed between the urban/palatial centre at Chania and the rural site at Hagios Charalambos, but no attempt is made to confirm this in other sites. Reference is made to Bintliff's (2017, 159) correlation between endogamy, social complexity and community size. However, Bintliff argues the opposite to what we observe at Chania – that endogamy is

9likely to arise when the size of a community increases beyond a set threshold. (It should perhaps be added that Bintliff's correlation is based on the kind of cross-cultural generalisations that characterized traditional evolutionary models now vehemently opposed by Graeber and Wengrow in *The Dawn of Everything*.) While the relation between demography and endogamy remains unclear, the discussion then attributes consanguinity to ecological factors which are not fully substantiated by the data, or cultural traditions which are not explored further. It is true that the relationship between mortuary practices and marital or residence rules in the past is new territory – but eventually geneticists (as well as prehistoric archaeologists) will need to delve more deeply into the theoretical aspects of kinship studies.

The results presented are of interest to scholars of (bio)archaeology, and the integrated approach adopted will increase its impact on the field.

We very much appreciate this constructive feedback and have completely restructured and reworded our discussion of the evidence of endogamy. We have removed those sentences in the Results section, which somehow anticipated the discussion and integrated them in the Discussion, where we have extended and reworded the last paragraph on endogamy. We have removed the reference to Bintliff and introduced reference to a few more anthropological studies (e.g., Lévi-Strauss). With regard to the relation between endogamy and demography, we have tried to make it clearer in our wording that the genetic evidence rather points to larger populations which means that small population size cannot be taken as an main reason for the importance of cousin-cousin unions. Whereas we were not able to expand in more detail on possible ecological and cultural factors for endogamy, we made clear that the diverse contexts (with regard to space, time, culture/society) where we could trace endogamy force us to take a complex bundle of reasons for this practice into account. We added, however, that the importance of olive in subsistence economies further enforced local constancy. We also made clear that future research will be necessary to unravel the complexity of factors and reasons behind this phenomenon. We also took the Reviewer's sentence "It is true...kinship studies" as an inspiration for our new last sentence of the discussion.

C. Data & methodology

Validity of approach

Many recent genetic studies brought important refinements in analytical or computational techniques, but sometimes offered flawed interpretations because they failed to place aDNA results in their context. The integrated approach adopted in this study is clearly a step in the right direction: The authors do not over-privilege genetic data, avoid simplistic or sensational conclusions, and allow nuance and alternative interpretations. As an archaeologist, I would like even more attention to context. To give just one example: lumping together the mountain plateau of Lasithi in Crete, the island of Salamis, and a site in Euboea, which mediates between the mainland and island culture, under 'islands' is questionable. In

10general, the authors should be more aware that the analytical categories employed in this paper - ‘the mainland’, ‘the islands’ and ‘Crete’ - intersect with the terms ‘Mycenaean’ or ‘Minoan’ which collapse geographic, chronological, cultural, ethnic or political designations and represent fluctuating networks of connectivity. See the remarks above (under B. Validity and significance) about part (ii) of the argument.

We are most pleased to read that our effort to bring together archaeology and archaeogenetics on equal level was appreciated by the reviewer, as this is at the heart of our thinking and approach. We agree that it is quite difficult to categorize different kinds of landscapes as our eagle eye-sight and modern-day perception might not (and most probably will not) do justice to past perceptions of space and landscapes. Therefore, we have removed those paragraphs where a classification into mainland vs. islands was too prominent and rather emphasize that phenomena like endogamy appear both, on the mainland as well as on what we classify as islands of different sizes. However, for our spatial categorization of genetic signatures we kept the category of “islands” vs. “mainland” out of practical reasons and the fact that some kind of categorization is necessary - and we had originally indeed thought of classifying Euboea as “mainland” but then decided to strictly keep modern geographical classifications - as arbitrary as they are. We make this now even clearer by adding the half-sentence in lines 273-274: “- being aware that such artificial subdivisions of landscapes might not reflect past categorizations.” We also removed any clear-cut association between cultural and geographic terms following the advice of the Reviewer.

Quality of data

The authors stress that they have used the largest genetic sample from the Aegean, and indeed this is an important achievement.

The archaeological data, with all their contextual differences, have been taken more into consideration when interpreting the data. I maintain that more needs to be done in this direction: more attention needs to be paid into differences of context (site size and status in the settlement hierarchy, age/gender or status of burial population, social structure and cultural traditions). This is meant more as a recommendation for future research and eventual integration and synthesis than as an amendment of this paper.

This recommendation is taken very seriously by us. Whereas this article is not the place to go in detail into the archaeological context, it is our focus for the next years to come. However, these integrative questions request a broader and more detailed bioarchaeological dataset and we believe that the dataset provided by us in our article is an important step towards enabling a comprehensive integrative approach in the future (which is the main topic of the ERC Consolidator Grant of P.W.S.).

Quality of presentation

The argument is built up in a clear and convincing way – although more so for part (i) than for part (ii) of the argument.

However, the end is abrupt and the two parts of the argument remain somewhat disjointed. They could be brought together more explicitly by stating that while the genetic history shows admixture and mobility from and to different directions, the postulated practice of consanguinity suggests, at least in some places and among some groups, a certain fixity in place and homogeneity in social practices across geographical, cultural, ethnic and political divides and through time. This could be done at the very end of the article.

We agree that in the last version the end of the article was rather abrupt. We have now completely rephrased our last paragraph and introduced a new last sentence inspired by one of the sentences of the reviewer. However, we had not the word space to expand in detail on the different geographical, cultural etc. factors, but hope that our new, more extensive last paragraph fits to the expectations of the reviewer.

D. Analytical approach

The statistical and computational methods employed fall outside the scope of my expertise.

E. Suggested improvements

The argument would improve

- a. If the authors would be more aware that the spatiotemporal categories they use in part (i) of the argument are artificial, and that the networks of connectivity within which mobility should be understood fluctuate in size and scope.
- b. If they would reflect more on part (ii) of the argument. it is advisable to avoid ecological explanations or vague references to cultural differences which have not been substantiated and to examine the theoretical premises of kinship structures more critically (the latter is meant more as recommendation for future research).

Both remarks can be addressed in the present paper by adjusting some formulations and without changing the scope of the argument.

We have now made very clear that our categories are artificial (and necessary from an epistemological point of view). We have also made clear (esp. in the case of endogamy) that we need to avoid simplistic explanations based on e.g. ecological or other factors but need to consider complex entanglements of reasons which we are just starting to understand. We have completely rewritten the last part of the discussion to take these recommendations of the reviewer better into account.

F. Clarity and context: lucidity of abstract/summary, appropriateness of abstract, introduction and

12conclusions

- The abstract is clear and accessible. However, the phrase “these transitions have been elucidated only to a limited extent” disregards the huge archaeological bibliography on these transitional periods. It should be replaced by something like “the role of human mobility, and specifically of genetic admixture, during these transitions only now begins to be studied in earnest.”
- The introduction is clear.
- See above (C. Quality of presentation) for a suggestion to improve the conclusions.

We have reworded the problematic sentence to: “...but for the Aegean ... the biological dimensions of cultural transitions have been elucidated only to a limited extent so far”, which makes clear that these limitations refer only to the biological understanding of past cultural transitions, and not to culture-historical studies.

G. References

Appropriate credit to previous work is given. It is good that references to studies which criticize biological determinism in genetic studies have been included (Brück 2021) although their message is still not fully internalized.

H. Minor remarks

- Why is Glyka Nera in the periphery of the Mycenaean world? (Supplementary Data, 1147-1148)?
- Clemente et al 2021 is not among the References

We reworded the sentence referring to Glyka Nera and removed any reference to “periphery”.

We forgot to include Clemente et al. 2021 in the references and have added the article now.

Sofia Voutsaki, Groningen

Reviewer #2 (Remarks to the Author):

Skourtanioti and colleagues have provided a thorough and thoughtful response to my comments. However, I have a few outstanding issues. These I believe are easily resolved and I'm happy to recommend for publication if addressed. I congratulate the authors on their excellent study, which I'm

13sure will draw much interest.

1. Methods and Data reporting

The authors briefly justify the lack of shotgun sequence to me in their response letter. However, it is important for the authors to provide these justifications to the reader in the main text, as well as the results from their initial shotgun screening in the supplementary tables. There are two reasons for this:

a) When destructive analysis is performed on archaeological material, there is an ethical imperative to get the most out of each individual sample. This is particularly true for time periods and geographical regions where DNA preservation is poor. As the authors note, data from only 30 ancient Aegean individuals was available prior to this study, and only a fraction of these are whole genome shotgun sequence. The authors had the opportunity to shotgun sequence at least seven of their samples (endogenous content >10%) to a coverage that would have more long-term value to the field but chose not to. I understand the logic underlying these decisions, but the authors should be up front with the reader about the trade-offs made and the limitations/advantages of capture data (e.g. the application of the same blanket approach to all samples, regardless of endogenous content, may help minimize batch effects).

We now address these points following the instructions from Reviewer 2 by adding additional explanations. One limitation is that we had to reduce the main text by ca. 600 words to comply with the NEE guidelines. But we now introduce in the subsection ‘The archaeogenetic dataset’ that only part of the immortalized libraries were enriched for the 1240K panel. Then we provided a more detailed explanation of our strategy in the Methods section (lines 636-649):

“Overall, our initial screening revealed that human ancient DNA preservation was overall very low to moderate (i.e., 0.1-10% human endogenous DNA). Therefore, only aDNA enrichment methods are an economically viable strategy that allows one to generate data from a large number of individuals. Here, we chose to minimize batch effects and consistently generated in-solution hybridization enrichment data, consisting of ~1,2 million ancestry-informative positions (‘1240K capture’) from all samples with 0.1% human endogenous DNA or more. We note that a small proportion of the sampled libraries exhibited high DNA preservation (9 samples with more than 10% and up to c. 40% endogenous content), that would make sequencing of the whole whole human genome cost-efficient, and doing so could address additional research questions (e.g., about rare variants). Only part of the immortalized libraries were used to

produce enrichment data. The remaining libraries are permanently stored at the MPI-SHH/EVA lab facilities and future studies can use this resource to generate whole-genome data from these libraries.”

b) The endogenous content results of the original shotgun screening may inform future genomic surveys of this region (e.g. which site types or geologies provide higher yields). Providing this data is also beneficial to the broader field as it provides important fodder for methodological improvement (which skeletal elements, extraction techniques etc. are providing best results).

Thank you for identifying this omission. We agree that these results of the initial screening are of interest for the broader field. We have updated Table S1 to include relevant statistics.

Also, could the authors provide two more minor clarifications:

a) In their response letter they inform me that:

“For the vast majority of the libraries with 1240k SNP data, we initially performed shotgun metagenomic sequencing (ca. 5M reads)”

However, in the main text they state that:

“All libraries were initially sequenced at a low depth (5-10M reads) on an Illumina HiSeq400 platform”

Which statement is correct – the vast majority or all?

Thank you for identifying this inconsistency in our reporting. We now corrected this line in the methods section:

“From every extract, at least one of the produced libraries was initially sequenced...” (lines 628-629).

b) The use of the term “endogenous DNA” in the supplementary material is confusing, as it usually refers to the endogenous DNA content of an extract or library prior to any targeting techniques. The results of capture are typically reported in terms of capture efficiency and numbers of on-target and off-target reads. As I am requesting the authors to report the true endogenous content of each library from the screening

15data, could the authors update the terminology used for the post-capture data to avoid confusion.

As suggested, we now include the endogenous DNA % of the screening data in Table S1 (see our reply to point 1b above). To indicate capture efficiency, we now also report the ‘number of on-target reads (%)’, both for shotgun and post-capture. Additionally, we report the number of 1240K SNPs covered at least once in Table S2 (as before).

2. Neolithic Affinities

I’m not quite satisfied with the authors response to my request for more dissection of Greek Neolithic affinities. Instead of carrying out further comparisons of Aposelemis_N and the Neolithic from the Peloponnese and Northern Mainland, they seem to have removed these two groups from most formal tests of admixture with the exception of Figure 3. For ease, I’ll refer to these populations as Mainland_N (excluding the youngest Peloponnese LN individuals with CHG-like ancestry likely derived from admixture in the 5th millennium BCE). This time around, I will be more direct with my requests for further analysis:

a) With regards to the PCA, perhaps my comments were not clear enough. Aposelemis_N does not plot with the Mainland_N samples. They differentiate on both PC1 and PC2 with W. Anatolians falling between the two groups. This may be driven by subtle differences in Levantine and Iranian ancestry. In Supp Table 4, Aposelemis_N shares a significant excess of alleles with Iran_C and near significant excesses with other populations high in Iranian_Neolithic ancestry. Why are the Mainland_N groups (S.Greece_EMNA and N.Greece_EN) not included in Supp Table 4? Do they show a different trend? Please add them and interpret the results.

We have now updated Table S4 to include these comparisons. Following your previous comment, we group the three individuals from the Early-Middle Neolithic Mainland (Rev5.SG, I5427 and I2937) as ‘Mainland_Greece_N’ and perform tests $f_4(\text{Mbuti, Test; W.Anatolia_N, Mainland_Greece_N})$, as well as $f_4(\text{Mbuti, Test; Mainland_Greece_N, Aposelemis_N})$. We want to highlight that Aposelemis_N and Mainland_Greece_N do overlap in the PCA with the individuals from W.Anatolia_N. It is only the later individuals (whitish square square) that do not, that is those requiring a two-way model as described in Figure 3. In addition to Figure 3, pairwise individual genetic affinities are presented in Figure S1 (pairwise *qpWave*).

b) Following from this, please perform direct comparisons between Aposelemis_N and Mainland_N using tests of the form $f_4(\text{Outgroup}, X_{\text{shotgun}}; \text{Aposelemis_N}, \text{Mainland_N})$ or $f_4(\text{Outgroup}, X_{\text{capture}}; \text{Aposelemis_N}, \text{Mainland_N_capture})$ to avoid any confounding effects of capture bias.

These tests are now added and comprise both shotgun and capture data at the second position (e.g., Iran_GanjDareh_N and Iran_TepeAbduHosein_N.SG). Similar trends in Z score values as with the test $f_4(\text{Mbuti}, \text{Test}; \text{W.Anatolia_N}, \text{Aposelemis_N})$ are observed, but not with $f_4(\text{Mbuti}, \text{Test}; \text{W.Anatolia_N}, \text{Mainland_Greece_N})$.

Because Aposelemis_N as a group has low heterozygosity (i.e., pairwise mismatch rate is equivalent to second-third degree relatives for the rest of the Aegean dataset - see also Figure S4), we now also provide the same f_4 tests but only on APO004, the individual with the highest SNP coverage. By using only one individual we might decrease the resolution, but all tests are still calculated on a high number of SNPs (c. 100,000-250,000). At the same time we can test for overestimation of the significance of the allele frequency differences between Aposelemis and W.Anatolia_N/ Mainland_Greece_N, owing to long-term inbreeding at Aposelemis. We note that the most positive Z scores decreased (now <2 , with the exception of Iran_C_TepeHissar, ANE and Levant_C with $3 < Z \leq 2$), while for some tests the sign became negative or the value more negative (i.e., WEHG, BalkanHG), making the tests more consistent with those from $f_4(\text{Mbuti}, \text{Test}; \text{W.Anatolia_N}, \text{Mainland_Greece_N})$. We also provide this explanation in Supplementary Note 2 (lines 12-13, 58-68)

c) Does a one-way model from Mainland_N also fit Aposelemis_N?

Yes! This information is now provided in Suppl. Note 2 (lines 229-231).

A minor comment: could the authors state within the main text that the genetic profile of the Mesolithic populations of the Aegean islands (Crete included) is unknown and it is possible that it was similar to that of Western Anatolian populations. There is no evidence of Villabruna-type ancestry any further southeast than the north Balkans (that I'm aware of?) and it possibly expanded from an Italian refugia. The general reader may not have this background context.

This point is significant, and we already raise it in the second paragraph of the discussion -now more clearly.

3. Minor corrections

Lines 205-207: This is hard to follow. Do you mean the earliest evidence of excess allele sharing with Neolithic Iran specifically is Nea Styra EBA? Or do you mean the earliest sample that gave a significant result for any test, which does not seem to be correct.

We meant the latter, and this seems to be the case: both Iranian Neolithic groups produce $Z < 3$ with both settings (on Aposelemis_N as a group or APO004).

We rephrase as follows:

“Affinities with far-eastern groups like Neolithic Iran are traced for Neolithic Aposelemis (or APO004), but only reach significance levels ($\geq 3SE$ or $Z \geq 3$) on the EBA group from Nea Styra, and then prevail for the majority of the later Aegean BA groups.” (lines 179-182).

Figure 1: The color and shape key for archaeological sites sampled is a little confusing. Why do both IA and LBA have the same circle symbol? If the authors require more shapes try out the ggstar package (<https://cran.r-project.org/web/packages/ggstar/vignettes/ggstar.html>). Also, it is not immediately clear that multiple points have been plotted for the same site and perhaps jittered on the map (e.g. Tirnys, Lazarides, Aposelemis). Are multiple points also plotted for published sites? For example, Diros Cave has both Early and Late Neolithic samples but is colored as Early. Perhaps clarify this in the legend and explain the color key also. It would be useful if published sites were included in the time line.

Thanks for pointing out these ambiguities and the new R-package!

Indeed, jitter has been applied for data dating to different periods but same locations. Now we specify this in the legend, and we also add jitter for the published individuals from the Neolithic site of Diros that date to different phases. We also changed the symbol for the Iron Age period, and updated the other figures. However, we kept the timeline as before, to provide the dating information from the new data with visual clarity.

Figure 3A: Please provide more of an explanation as to the meaning of the arrows and dotted axes placed below the main ternary plot. I think you are separating out Crete and Mainland samples along the Anatolia-Iran axis but it is not quite clear to reader.

Yes, this is right. We now clarify this in the figure legend.

Reviewer #3 (Remarks to the Author):

The authors have addressed in a satisfactory way virtually all the points I have raised in my previous evaluation of tehri manuscript, and I do not have any other major concern beside one.

We thank Reviewer 3 for the positive feedback!

Authors have mentioned in the text that HG.Charalambos have many related individuals. Maybe authors should explore how this increased relatedness is affecting the analyses based on this dataset as related individuals would be removed if the analysis would be done on modern day populations. Given the large set of individuals (n=28) I suspect that after pruning the sample size would be numerically not too different from the size of other groups they analysed (mostly in ranging between 1 and 9)

We had already removed closely related individuals from the Hg.Charalambos dataset at the group-based analyses as indicated in Table S2. This reduced the dataset to 22 individuals.

Minor concerns

In relation to the variation in Iran/Caucasus-related ancestry in the reply (and in teh ms) the authors mention that "male exogamy could also contribute to this pattern". It might be relevant here to verify if their Y chromosome/X chromosome data are compatible with this scenario. Whatever the situation (yes, or not/inconclusive), this should be mentioned.

We totally agree with this point. However, even after Y chromosome capture, it was not possible to resolve the haplogroups/haplotypes for the individuals from Nea Styra. Therefore, we now simply added: "...although Y-haplogroups are unresolved,..." (line 386).

In answering my question about relationship between sample coverage and ROH authors refer to the methods and its development; my question was specific to this dataset: is there any relation between coverage and ROH in this dataset? Again, authors should include an appropriate comment in the ms.

We are sorry that we did not understand your point from the beginning. Because we cannot add more text -rather the contrary- we address your comment with an additional panel at Figure 6 (C). The plot is limited only to the range 250,000-400,000 SNPs, which represents the new lower threshold applied here . The correlation does not appear strong, we briefly comment that in one phrase at the legend of Figure 6.

When referring to the occurrence of cousin-cousin marriage authors refer to a wikipedia page to support their response. I would suggest to check the robustness of this by looking at more classical anthropological surveys and refer directly to them (as for example the Atlas of World Cultures by Murdock), also in the main text.

To our knowledge, no wikipedia page has been cited. However, following also the comments from reviewer 1, we have rephrased this part (last paragraph of Discussion), adding more relevant citations (e.g., the fundamental work from Lévi-Strauss).

Faced with the options presented by authors, for clarity I would suggest to add a supplementary figure (in response to my comment on Figure 1 clarity).

We forgot to clarify previously that Figure S3.2 (embedded in Supplementary Note 3) is a PCA built in the exact same way as Figure 2, but also includes the three-letter annotation of every modern population plotted as the mean of PC1-PC2 coordinates from all the individuals.

I would suggest to split the Results section "Trans-regional genetic entanglements of Aegean populations" in subsections to make it more readable. AT the moment is almost four pages long; maybe a smaller section on Neolithic (which show little differences) and 2 on Bronze Age, maybe comparing continental/Crete results?

In the current version, parts of methodological details have been removed from the results, making the section shorter. Following your recommendations, we now created the subsections "Neolithic-Early/Middle Bronze Age", and "Mobility in the Middle/Late Bronze Age Aegean".

229-230: provide errors associated with these estimates

Added (lines 202-203).

Occurrence of Y chromosome types (324-326): Occurrence of J and G chromosomes in Iran, Caucasus, Anatolia and Levant compatible with their other data linking these samples with these areas. Were these haplogroups present in other regions too and therefore similarly compatible?

We rephrased this part adding that these haplogroups have also been common throughout Neolithic Europe (lines 295-296).

Decision Letter, first revision:

5th September 2022

Dear Eirini,

Thank you for submitting your revised manuscript "Ancient DNA reveals admixture history and endogamy in the prehistoric Aegean" (NATECOLEVOL-220516439A). It has now been seen again by the original reviewers and their comments are below. The reviewers find that the paper has improved in revision, and therefore we'll be happy in principle to publish it in Nature Ecology & Evolution, pending minor revisions to comply with our editorial and formatting guidelines.

[REDACTED]

Reviewer #1 (Remarks to the Author):

I have now looked at the revised manuscript, and I recommend it for publication in Nature Ecology & Evolution.

21Reviewer #2 (Remarks to the Author):

I'm happy with the authors responses and have no further comments. Congratulations to all again on the great study!

Reviewer #3 (Remarks to the Author):

The authors have addressed the issues I raised in my previous review. I congratulate them for the extensive rewriting and re-organization of the main text that significantly improved the readability and accessibility to their work.

Our ref: NATECOLEVOL-220516439A

8th September 2022

Dear Dr. Skourtanioti,

Thank you for your patience as we've prepared the guidelines for final submission of your Nature Ecology & Evolution manuscript, "Ancient DNA reveals admixture history and endogamy in the prehistoric Aegean" (NATECOLEVOL-220516439A). Please carefully follow the step-by-step instructions provided in the attached file, and add a response in each row of the table to indicate the changes that you have made. Please also check and comment on any additional marked-up edits we have proposed within the text. Ensuring that each point is addressed will help to ensure that your revised manuscript can be swiftly handed over to our production team.

****We would like to start working on your revised paper, with all of the requested files and forms, as soon as possible (preferably within two weeks). Please get in contact with us immediately if you anticipate it taking more than two weeks to submit these revised files.****

In recognition of the time and expertise our reviewers provide to Nature Ecology & Evolution's editorial process, we would like to formally acknowledge their contribution to the external peer review of your

22manuscript entitled "Ancient DNA reveals admixture history and endogamy in the prehistoric Aegean". For those reviewers who give their assent, we will be publishing their names alongside the published article.

Nature Ecology & Evolution offers a Transparent Peer Review option for new original research manuscripts submitted after December 1st, 2019. As part of this initiative, we encourage our authors to support increased transparency into the peer review process by agreeing to have the reviewer comments, author rebuttal letters, and editorial decision letters published as a Supplementary item. When you submit your final files please clearly state in your cover letter whether or not you would like to participate in this initiative. Please note that failure to state your preference will result in delays in accepting your manuscript for publication.

Cover suggestions

As you prepare your final files we encourage you to consider whether you have any images or illustrations that may be appropriate for use on the cover of Nature Ecology & Evolution.

Nature Ecology & Evolution has now transitioned to a unified Rights Collection system which will allow our Author Services team to quickly and easily collect the rights and permissions required to publish your work. Approximately 10 days after your paper is formally accepted, you will receive an email in providing you with a link to complete the grant of rights. If your paper is eligible for Open Access, our Author Services team will also be in touch regarding any additional information that may be required to arrange payment for your article.

Please note that *Nature Ecology & Evolution* is a Transformative Journal (TJ). Authors may publish their research with us through the traditional subscription access route or make their paper immediately open access through payment of an article-processing charge (APC). Authors will not be required to make a final decision about access to their article until it has been accepted. [Find out more about Transformative Journals](https://www.springernature.com/gp/open-research/transformative-journals)

Authors may need to take specific actions to achieve [compliance](https://www.springernature.com/gp/open-research/funding/policy-compliance-faqs) with funder and institutional open access mandates. If your research is supported by a funder that requires immediate open access (e.g. according to [Plan S principles](https://www.springernature.com/gp/open-research/plan-s-compliance)) then you should select the gold OA route, and we will direct you to the compliant route where possible. For authors selecting the subscription publication route, the journal's standard licensing terms will need to be accepted, including [self-archiving-and-license-to-publish](https://www.nature.com/nature-portfolio/editorial-policies/self-archiving-and-license-to-publish). Those licensing terms will supersede any other terms that the author or any third party may assert apply to any version of the manuscript.

[REDACTED]

[REDACTED]

Reviewer #1:

Remarks to the Author:

I have now looked at the revised manuscript, and I recommend it for publication in Nature Ecology & Evolution.

Reviewer #2:

Remarks to the Author:

I'm happy with the authors responses and have no further comments. Congratulations to all again on the great study!

Reviewer #3:

24Remarks to the Author:

The authors have addressed the issues I raised in my previous review. I congratulate them for the extensive rewriting and re-organization of the main text that significantly improved the readability and accessibility to their work.

Final Decision Letter:

11th November 2022

Dear Eirini,

We are pleased to inform you that your Article entitled "Ancient DNA reveals admixture history and endogamy in the prehistoric Aegean", has now been accepted for publication in Nature Ecology & Evolution.

Over the next few weeks, your paper will be copyedited to ensure that it conforms to Nature Ecology and Evolution style. Once your paper is typeset, you will receive an email with a link to choose the appropriate publishing options for your paper and our Author Services team will be in touch regarding any additional information that may be required

You will not receive your proofs until the publishing agreement has been received through our system

Due to the importance of these deadlines, we ask you please us know now whether you will be difficult to contact over the next month. If this is the case, we ask you provide us with the contact information (email, phone and fax) of someone who will be able to check the proofs on your behalf, and who will be available to address any last-minute problems . Once your paper has been scheduled for online publication, the Nature press office will be in touch to confirm the details.

Acceptance of your manuscript is conditional on all authors' agreement with our publication policies (see www.nature.com/authors/policies/index.html). In particular your manuscript must not be published elsewhere and there must be no announcement of the work to any media outlet until the publication date (the day on which it is uploaded onto our web site).

Please note that *Nature Ecology & Evolution* is a Transformative Journal (TJ). Authors may publish their research with us through the traditional subscription access route or make their paper immediately open access through payment of an article-processing charge (APC). Authors will not be

25required to make a final decision about access to their article until it has been accepted. [Find out more about Transformative Journals](https://www.springernature.com/gp/open-research/transformative-journals)

Authors may need to take specific actions to achieve [compliance with funder and institutional open access mandates](https://www.springernature.com/gp/open-research/funding/policy-compliance-faqs). If your research is supported by a funder that requires immediate open access (e.g. according to [Plan S principles](https://www.springernature.com/gp/open-research/plan-s-compliance)) then you should select the gold OA route, and we will direct you to the compliant route where possible. For authors selecting the subscription publication route, the journal's standard licensing terms will need to be accepted, including [editorial policies/self-archiving-and-license-to-publish](https://www.nature.com/nature-portfolio/editorial-policies/self-archiving-and-license-to-publish). Those licensing terms will supersede any other terms that the author or any third party may assert apply to any version of the manuscript.

We welcome the submission of potential cover material (including a short caption of around 40 words) related to your manuscript; suggestions should be sent to Nature Ecology & Evolution as electronic files (the image should be 300 dpi at 210 x 297 mm in either TIFF or JPEG format). Please note that such pictures should be selected more for their aesthetic appeal than for their scientific content, and that colour images work better than black and white or grayscale images. Please do not try to design a cover with the Nature Ecology & Evolution logo etc., and please do not submit composites of images related to your work. I am sure you will understand that we cannot make any promise as to whether any of your suggestions might be selected for the cover of the journal.

To assist our authors in disseminating their research to the broader community, our SharedIt initiative provides you with a unique shareable link that will allow anyone (with or without a subscription) to

26read the published article. Recipients of the link with a subscription will also be able to download and print the PDF.

You can generate the link yourself when you receive your article DOI by entering it here: <http://authors.springernature.com/share>.

[REDACTED]

P.S. Click on the following link if you would like to recommend Nature Ecology & Evolution to your librarian <http://www.nature.com/subscriptions/recommend.html#forms>

** Visit the Springer Nature Editorial and Publishing website at http://editorial-jobs.springernature.com?utm_source=ejp_NEcoE_email&utm_medium=ejp_NEcoE_email&utm_campaign=ejp_NEcoE for more information about our career opportunities. If you have any questions please click [here](mailto:editorial.publishing.jobs@springernature.com). **